# Synthetic intrinsically disordered protein fusion tags that enhance protein solubility

Nicholas C. Tang[1,2], Jonathan C. Su[1,2], Yulia Shmidov [1,2], Garrett Kelly[1], Sonal Deshpande[1], Parul Sirohi[1], Nikhil Peterson [1] & Ashutosh Chilkoti [1] ✉

We report the de novo design of small (<20 kDa) and highly soluble synthetic intrinsically disordered proteins (SynIDPs) that confer solubility to a fusion partner with minimal effect on the activity of the fused protein. To identify highly soluble SynIDPs, we create a pooled gene-library utilizing a one-pot gene synthesis technology to create a large library of repetitive genes that encode SynIDPs. We identify three small (<20 kDa) and highly soluble SynIDPs from this gene library that lack secondary structure and have high solvation. Recombinant fusion of these SynIDPs to three known inclusion body forming proteins rescue their soluble expression and do not impede the activity of the fusion partner, thereby eliminating the need for removal of the SynIDP tag. These findings highlight the utility of SynIDPs as solubility tags, as they promote the soluble expression of proteins in *E. coli* and are small, unstructured proteins that minimally interfere with the biological activity of the fused protein.

The overexpression of many recombinant proteins can lead to improper folding, in vivo degradation, or their sequestration in insoluble inclusion bodies, all of which result in low yields of soluble and functionally active protein[1,2]. One solution to this problem is to use a fusion partner to enhance the solubility and activity of a target protein[3]. However, most proteins that are used as fusion tags to enhance the soluble expression of recombinant proteins, such as maltose binding protein (MBP)[4–8], thioredoxin[7,9,10], Nus A[8,11], or small ubiquitin related modifier (SUMO)[12] were empirically identified and were not designed by nature or evolved for this purpose. Furthermore, these tags promote solubility by preventing aggregation, but may not always facilitate proper folding and can result in soluble inactive proteins[7,8]. Hence, there is a clear need to design peptide-based tags de novo that facilitate the overexpression of soluble, folded and functionally active proteins.

It has been previously shown that disordered regions at the N terminus of folded native proteins promote solubility and prevent aggregation of proteins[13,14]. Due to lack of secondary structure, these disordered regions have high conformational freedom, and hence are believed to act as entropic bristles which prevent aggregation of the protein. Based on this knowledge, we hypothesized that polypeptide tags derived from intrinsically disordered proteins (IDPs) can promote the soluble expression of recombinant proteins.

To explore this hypothesis, we report the de novo design of small (<20 kDa) and highly soluble SynIDPs that confer solubility to a fusion partner with minimal effect on the bioactivity of the fused protein[15]. To identify highly soluble SynIDPs, we create a pooled gene-library to create a large library of repetitive genes that encode SynIDPs. We transform this pooled gene library into *E. coli* and identify unique clones that show a high level of soluble expression of the SynIDP. We then demonstrate that the fusion of three highly soluble SynIDPs identified from this pipeline improves the solubility of three proteins that are notoriously difficult to produce as soluble and functionally active proteins in *E. coli* due to their aggregation into insoluble inclusion bodies: (1) murine terminal deoxynucleotidyl transferase (mTdT)[16,17]; (2) a modular fusion protein of a dimeric affibody domain (Z2) that targets epidermal growth factor receptor (EGFR)[18] fused to a *P. aeruginosa exotoxin A* (Z2-LO10)[19]; and (3) tobacco etch virus (TEV) protease[8,20,21]. Finally, we show that the activity of each protein is maintained even without cleaving off the SynIDP tags, demonstrating that the tag promotes soluble expression and folding and does not interfere with the activity of its fusion partner.

[1]Department of Biomedical Engineering, Duke University, Durham, NC 27708, USA. [2]These authors contributed equally: Nicholas C. Tang, Jonathan C. Su, Yulia Shmidov. ✉e-mail: chilkoti@duke.edu

## Results

### Native IDPs inspire a diverse set of SynIDPs

Native IDPs are highly dynamic, as they have significant stretches of amino acids that lack a defined tertiary structure and have a high degree of solvent-exposure[22,23]. These properties of IDPs can enable folding of their fused protein partner, and thereby promote soluble protein expression[24–26]. However, native IDPs have several potential limitations as tags for the recombinant expression of proteins. First, native IDPs come in different sizes, and the extent of their intrinsically disordered regions—IDRs—span a wide range of their primary sequence[27]. Second, native IDPs have a specific cellular function, so that their recombinant overexpression in cells could potentially interfere with cellular function or metabolism[28]. Hence, we decided to create SynIDPs in the size range of 10–20 kDa that are completely disordered, motivated by the hypothesis that a complete lack of secondary structure would impart the highest possible degree of solvent exposure, and is also unlikely to impart—potentially interfering—biological function to these SynIDPs. We chose the 10–20 kDa size range since in our experience, peptides smaller than 10 kDa are poorly expressed, making them difficult to characterize, while proteins larger than 20 kDa can cause metabolic stress in cells overexpressing fusion proteins. In addition, in a previous study exploring disordered proteins as solubility tags, a 15 kDa disordered tag was identified as one of the most effective[25]. We hypothesized that these SynIDPs may promote the solubility of proteins that they are fused to, and therefore may serve as a useful tag for the soluble expression of proteins that are known to form inclusion bodies in *E. coli*.

To design SynIDPs that have these attributes, we decided to create SynIDPs that consist of repeats of short sequences of low complexity (SLCs) that are prevalent in native IDPs[29,30]. Our design was informed by our previous work on the identification of sequence heuristics to design repetitive polypeptides of SLCs that are intrinsically disordered[31]. These proteins consist of repeats of a P-G-$X_n$ SLC sequence, where *n* varies from 0 to 4. The periodic Pro and Gly residues were chosen as they are structure-breaking residues that are ubiquitous in naturally occurring IDPs[32] and the X residues represent a diverse set of amino acids. Repetitive polypeptides of these SLC sequences are the simplest possible—minimal—SynIDPs in terms of their sequence complexity.

We hypothesized that in the enormous set of $>20^4$ possible sequences of these SynIDPs, there must exist a subset of highly soluble SynIDPs. To test this hypothesis, we next describe the gene synthesis of a library of SynIDPs, their expression, and the identification of a subset of SynIDPs that show a high level of soluble expression in *E. coli*.

### P-G-$X_n$ gene libraries generate SynIDPs with high diversity

We selected a highly diverse set of 1020, 72-nucleotide (nt) long sequences that encode $(PGX_1X_2X_3X_4)_4$, where $X_1$ through $X_4$ represent any amino acid except proline and cysteine, and glycine occurs at least once in each motif (Supplementary Data 1, Supplementary Data 2). Cysteine was excluded due to its propensity to form disulfide bridges that are implicated in protein aggregation in bacterial expression systems[33,34]. We excluded glycine from $X_4$ to limit the presence of GP dipeptides which have been shown to promote irreversible aggregation[35]. X is chosen to be glycine at least once per repeat, as it is a small hydrophilic residue that imparts conformational flexibility to the SynIDP[36]. In this repeat motif, the proline content is ~15% and the glycine content is ~30%, which is consistent with the composition of highly disordered native elastomeric domains that these SLCs are inspired by[37]. With 1020 sequences, all possible selections of amino acids for $X_n$ except proline and cysteine are represented, ignoring permutations (Supplementary Data 1).

We designed the 72-nt long sequences to contain exactly one non-palindromic methylation-sensitive SexAI recognition site to overlap with the Pro-Gly coding sequence (Supplementary Fig. 1). Codons were chosen using the codon scrambling algorithm that we had previously developed to remove inverted nucleotide repeats or hairpins to maximize success of subsequent reactions with DNA polymerase or ligase[38].

### Rolling circle amplification (RCA) generates a pooled SynIDP gene library

We developed a method for the pooled, "one-pot" synthesis of repetitive gene libraries from the oligonucleotides on the order of one gene per oligonucleotide molecule using multiply primed RCA (Fig. 1A)[39]. RCA is preferable over traditional polymerase chain reaction (PCR) to generate repetitive DNA sequences due to potential misalignments in traditional PCR primer annealing, and difficulties in in vitro gene synthesis arising from secondary structure formation from repetitive sequences[40]. The entire pool of 1020 linear 72-nt long ssDNA molecules was circularized via CircLigase II (Epicenter (Lucigen), USA). Non-circularized DNA was removed by incubating ligated products with Exonuclease I and Exonuclease III. An exonuclease resistant primer with 3'-terminal phosphorothioate (PTO) modifications (Thermo Fisher Scientific) was annealed to the circular templates prior to isothermal polymerization at 30 °C using φ29 DNA polymerase, a highly processive and strand displacing polymerase. 5-methylcytosine triphosphate (5mCTP) was added to the dNTP mix to introduce the modified nucleotide approximately every four repeats (where one repeat is 72 nucleotides). In multiple primed RCA, a cascade of random priming events generates higher molecular weight double stranded DNA from each individual template simultaneously, as the random primer can anneal to the ssDNA product and the polymerase can generate a complementary strand.

The resulting products were digested by SexAI, a methylation sensitive enzyme with activity blocked by dcm methylated restriction sites. As the SexAI cut site was designed to overlap with the Pro-Gly coding sequence and was split to the 5' and 3' ends of the oligonucleotide sequence, all digested sequences would have a $(GX_1X_2X_3X_4P)_n$ sequence. Additionally, ssDNA sequences would not undergo digestion due to the enzyme's specificity for dsDNA (Supplementary Fig. 1). Digested fragments were separated by gel electrophoresis and exhibited a ladder pattern due to random methyl-C incorporation (Fig. 1B, right lane). We isolated bands by gel purification between 400 and 600 nucleotides, corresponding to 5-8 repeats of the 72 nt long template that encode polypeptides with 22-28 repeats of the $PGX_1X_2X_3X_4$ motif. The purified DNA was ligated using ElectroLigase (NEB) into a pET-24a expression vector that is modified to encode a MSKGP sequence at the N-terminus of the SynIDP with a seamless SexAI recognition site followed by a DNA sequence that encodes a C-terminal ENLYFQG-$(H)_6$ peptide where $H_6$ denotes the $(His)_6$ tag and the ENLYFQG peptide sequence (referred to hereafter as tev) is the substrate-cleavage site of TEV protease. The $H_6$ tag enables purification of the SynIDPs by immobilized metal affinity chromatography (IMAC) and the cleavage site allows cleavage of the protein from the SynIDP tag.

The plasmids with SynIDP inserts were electroporated into DH10B electrocompetent cells and plated on kanamycin agar plates. Following overnight growth, colonies were collected by scraping with 1 mL of fresh media, and the plasmid DNA were purified by a plasmid isolation kit in a single tube, and prepared for next generation sequencing (NGS, see experimental).

### Next generation sequencing (NGS) verifies library diversity

Sequence patterns and diversity were verified by NGS of the genes, using a 2 × 251 bp paired end flow cell in the Illumina Miseq NGS system. The sequences were aligned using a Burrows-Wheeler Aligner (BWA)[41], where the outputs were mapped against the theoretical 1020 reference sequences within the sequence space. Out of 2 million sequences returned by NGS, there were 3888 unique alignments based

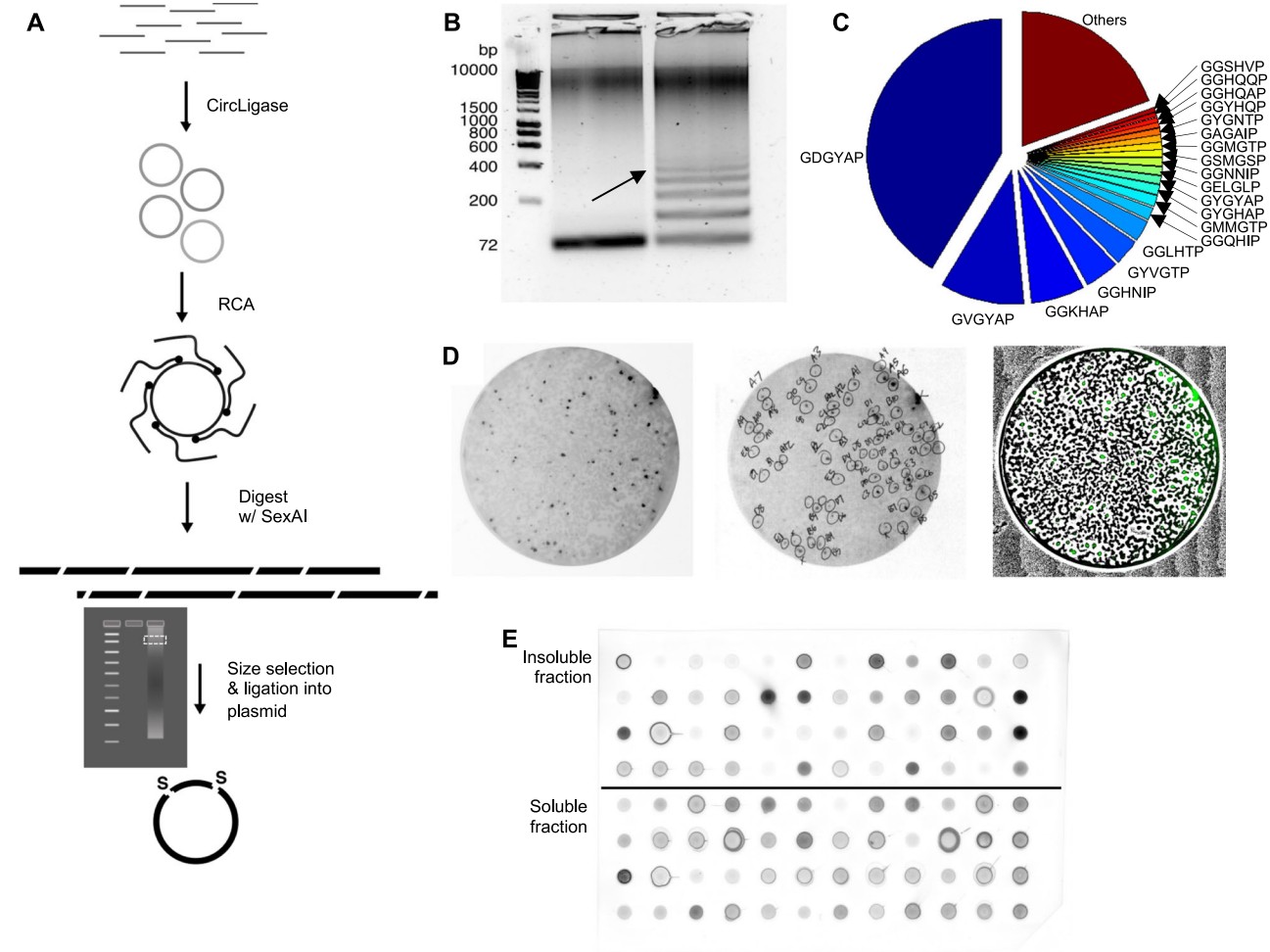

**Fig. 1 | The steps for identification of highly soluble SynIDPs. A** Generation of SynIDP gene library. A pool of 1020 ssDNA 72-nt long oligonucleotides was circularized by CircLigase and amplified using RCA. Substitution of dCTP with 10% 5-methyl-dCTP randomly incorporates 5mC into the RCA product, resulting in various DNA size products upon digestion by SexAI (see Supplementary Fig. 1). The products are separated by electrophoresis. Fragments of the desired size (360–576 bp) are then gel purified and ligated into a plasmid. **B** Restriction digestion of RCA products with 0% 5mCTP (left) vs. 10% 5mCTP (right) using the restriction enzyme SexAI shows that digests occur in multiples of 72 bp when 5mC is incorporated. Desired size (360–576 bp, marked with arrow) DNA are then gel purified (**C**). Illumina Miseq analysis of cloned library of plasmid DNA verifies the presence of 865 unique motifs (See Supplementary Fig. 2 for enlarged version). Source data are provided as a Source Data file (**D**). CoFi protein expression

(see Supplementary Fig. 3) allows identification of *E. coli* colonies that express soluble protein by an anti-His$_6$ western blot (left panel). Colonies were identified manually on a greyscale image (middle panel) and a black and white pixel-thresholded image overlaying the green channel (right panel), was printed and aligned underneath the plate of colonies as a visual aid for colony picking. Images represent a subset of the total set of agar plates that were analyzed (see Supplementary Fig). **E** Dot blot of insoluble and soluble lysate fractions to determine target protein solubility. The dot blot membrane was subjected to His$_6$-tag antibody detection. The distinct dots marked by arrows represent an example target SynIDP–[GTHGTP]$_{24}$–that was determined to be soluble. Solubility was determined by the intensity of the soluble fraction relative to the intensity of the insoluble fraction in both dot blot and PAGE gels (see Supplementary Figs. 6, 7).

on reference oligonucleotides paired with compact idiosyncratic gapped alignment reports strings (Fig. 1C, Supplementary Fig. 2). Alignments were then filtered for sequences that were too short, aligned poorly, or contained nonsense mutations, large frameshifts, or less than 3 perfect repeats. Elimination of these sequence left 865 unique sequences, which included perfect alignments and sequences with substitutions or small frameshifted regions. These remaining sequences aligned with 419 of the original 1020 oligonucleotides due to the introduction of sequence bias. Sequence bias could arise from a variety of factors including, but not limited to, polymerase nucleotide insertion preference, preferential plasmid replication in vivo and toxicity. An additional 180 unique sequences aligned with reverse complements of the oligonucleotide references, resulting from cloning of a fragment in the opposite orientation, but still resulting in a proline/glycine rich amino acid sequence.

## Colony filtration (CoFi) screens for soluble SynIDP expression

To rapidly identify soluble protein expression in *E. coli*, we implemented the Colony Filtration (CoFi) blot method (see experimental, Supplementary Fig. 3)[42]. Briefly, a mix of plasmid DNA was electroporated into Acella™ electrocompetent BL21(DE3) *E. coli* and plated on LB-kanamycin plates. Plates were replica plated[43] on additional LB-kanamycin plates prior to transfer of the original plate's colonies on to a nitrocellulose membrane over a filter sandwich consisting of a Whatman paper drenched in lysis buffer. The resulting soluble blots were visualized by Western Blot with an anti-His$_6$ antibody (Fig. 1D, Supplementary Fig. 4).

The CoFi method positively screens for soluble proteins at the colony level. Dark spots on a His$_6$-tag Western blot indicate colonies expressing soluble full-length proteins (left panel, Fig. 1D). These dark spots were manually identified for colony-selection from a greyscale

image that was printed on paper (middle panel, Fig. 1D). To facilitate the visual identification of previously selected colonies on the replica plates, the greyscale image was processed (right panel, Fig. 1D). The resulting color image was properly scaled, printed, and fiducially aligned underneath the semi-transparent agar replica plates, facilitating subsequent colony picking. In this way, candidate SynIDPs for subsequent analysis were selected by their soluble expression level in CoFi.

## Colony PCR identifies undesired plasmids

Next, we used PCR amplification directly from the *E. coli* colonies that showed soluble expression in CoFi to screen and eliminate from further consideration short genes of undesired lengths that are not removed by the CoFi method. Short genes of suboptimal lengths were likely isolated with the longer genes during the size selection process due to aberrations in gel migration and/or cross-hybridization between different DNA molecules. For particularly dense plates, some colonies contained mixtures of genes due to cross-contamination of neighboring colonies. Mixed populations of genotypes are less clearly identified by Sanger sequencing and must be identified by the colony-PCR screen to be eliminated from further consideration[44,45] (Supplementary Fig. 5).

Following colony screening, 572 colonies were sent to a Sanger sequencing vendor (Azenta) for direct DNA sequencing from *E. coli* cultures. At this stage, duplicates of different lengths were removed, and the remaining colonies were labeled (Fig. 1D middle panel) so that their sequences could be tracked during protein expression. A total of 52 parent motifs (Supplementary Table 1) were conserved, where some sequences included minor missense mutations, which slightly changes the overall amino acid composition of the parent motif. For each of these 52 parent motifs, we selected a unique sequence with greater than 20 repeats of the monomer for expression.

## SynIDP library expression identifies soluble clones

*E. coli* BL21(DE3) colonies containing plasmids encoding 52 unique SynIDP sequences were grown in liquid TB autoinduction media to express each recombinant protein. To facilitate high throughput protein production, we used the Duetz Microflask system to cultivate cells with high growth rates and protein yields. Following growth of the *E. coli* BL21(DE3) cells harboring a SynIDP encoding plasmid for 24 h, clones were then re-sequenced to further confirm the absence of deleterious frameshifts or mixed populations, and the duplicates were then removed leaving single representative clones. Cells from the resulting clones were harvested for purification. Soluble and insoluble fractions of the cell lysate were screened for protein expression. Insoluble fractions were dissolved in 8 M urea, and both fractions were run on an SDS-PAGE gel (Supplementary Fig. 6). Both fractions were also blotted onto a nitrocellulose membrane to quantify the relative amounts of soluble and insoluble fractions by an anti-His$_6$ western blot.

Visualization of the blot assay showed distinct dots (Fig. 1E, Supplementary Fig. 7) for the soluble and insoluble fractions. Solubility was determined by the intensity of soluble fraction relative to intensity of the insoluble fraction in both dot blot and PAGE gels. Candidate SynIDPs for subsequent analysis were selected by the following criteria: (1) Soluble expression level in CoFi; (2) Soluble expression level in dot blot and PAGE gels; (3) Homogenous DNA plasmid population.

Based on the criteria above, we selected a subset of five SynIDPs that exhibited the highest soluble expression and evaluated their ability to confer solubility to fusion protein: [GQSGLP]$_{24}$, [GTHGTP]$_{24}$, [GIGQAP]$_{20}$, [GANMPQ]$_{24}$, and [GAGAIP]$_{24}$. [GANMPQ]$_{24}$ and [GAGAIP]$_{24}$ have missense mutations—noted by the underlined residues— and have the sequence [GANMPQGAS̲IPPGANI̲PPGASI̲PP]$_6$ and [GAGAIPGAE̲AIPGAGAIPGAGAIP]$_6$ respectively.

## Characterization of SynIDPs demonstrates lack of secondary structure but different peptide chain properties

To examine the utility of these five SynIDPs as solubility tags, we surveyed the literature for proteins with a useful biological function that are known to express insolubly in *E. coli*, but not due to incorrect disulfide bond formation. The three proteins we chose—mTdT, Z2-LO10, and TEV protease—satisfy these criteria[16,19,21] (Supplementary Fig. 8, Supplementary Table 2). These proteins also have an easy measure of functional activity as two of them—mTdT and TEV protease —are enzymes, while Z2-LO10's function can be measured through metabolic arrest, leading to cellular death of epidermal growth factor receptor (EGFR)-expressing cells[46,47]. These proteins are also of interest in biotechnology and medicine. Z2-LO10 is a modular biologic drug for cancer treatment that consists of a dimeric affibody targeting domain and a potent bacterial toxin payload[18,48–50]; TEV protease is commonly used to cleave recombinant proteins from their tags[51]; and TdT is a reagent for the TUNEL apoptosis assay and for the de novo synthesis of DNA[17,52,53].

SDS-PAGE of insoluble and soluble fractions of these proteins (Supplementary Fig. 8A) expressed in BL21(DE3) *E. coli* at 37 °C showed that TEV and Z2-LO10 are indeed insoluble. However, the expression of mTdT at 37 °C is low, even as insoluble protein. As previous studies have reported high expression levels of mTdT at lower temperatures[54], we attempted its expression at 16 °C. At these lower temperatures, there is a high level of expression of mTdT, although most of it is still insoluble (Supplementary Fig. 8B).

As there is significant interest in using TdT for biotechnology applications[17,53,55–60], there is critical need to develop new methods for the large scale and economical expression of highly processive TdT as a reagent for this emerging enzymatic DNA synthesis technology. For these reasons, we chose mTdT as the first protein to test the ability of the SynIDPs to promote soluble expression. The genes encoding the SynIDP-mTdT fusion proteins were constructed using Gibson assembly by appending the mTdT gene to the C-terminus of each of the five SynIDP-tev-H$_6$ genes[61,62]. The fusion genes were inserted into a pET-24 expression vector and were transformed into BL21(DE3) *E. coli*. The transformed cells were grown overnight using overnight express TB. Western blot analysis of the soluble fraction of cell lysate (Supplementary Fig. 9) using an anti-His antibody showed soluble expression of three SynIDP-tev-H$_6$-mTdT fusions: [GQSGLP]$_{24}$-tev-H$_6$-mTdT that we hereafter refer to as SynIDP-1-mTdT, [GTHGTP]$_{24}$-tev-H$_6$-mTdT named SynIDP-2-mTdT, and [GAGAIP]$_{24}$-tev-H$_6$-mTdT named SynIDP-3-mTdT. In comparison, the expression of H$_6$-mTdT without a solubility tag resulted in an undetectable level of protein in the soluble fraction of the cell lysate (Supplementary Fig. 9).

Having confirmed that these three SynIDPs (Supplementary Table 3) promote the soluble expression of mTdT, we next sought to study their physicochemical properties and their structure. To do so, we expressed SynIDPs-tev-H$_6$ in *E. coli* BL21(DE3) and purified the proteins by IMAC (Supplementary Fig. 10A) followed by size exclusion chromatography (SEC). SynIDP-2-tev-H$_6$ has histidine residues in every repeat unit and hence requires a higher imidazole concentration to elute than the two other SynIDP fusions, resulting in purer protein than the other two SynIDPs. SDS-PAGE of the purified SynIDPs (Fig. 2A) shows that the SynIDPs migrate at a larger molecular weight than expected for globular proteins, which has been observed for other SynIDPs[63]. To verify the mass of the SynIDPs, mass spectrometry (MS) analysis was performed. For all three SynIDPs, the experimentally measured mass by matrix assisted laser desorption ionization time-of-flight (MALDI-TOF) MS (Supplementary Fig. 11) exactly matched their predicted mass (Supplementary Table 3). However, the MALDI-TOFMS spectrum of SynIDP-2-tev-H$_6$ has an additional peak that is consistent with a protein that is truncated within the tev−ENLYFQG−site.

Circular dichroism (CD) spectroscopy at 37 °C (Fig. 2B) showed that the SynIDPs lack defined secondary structure, as they have a

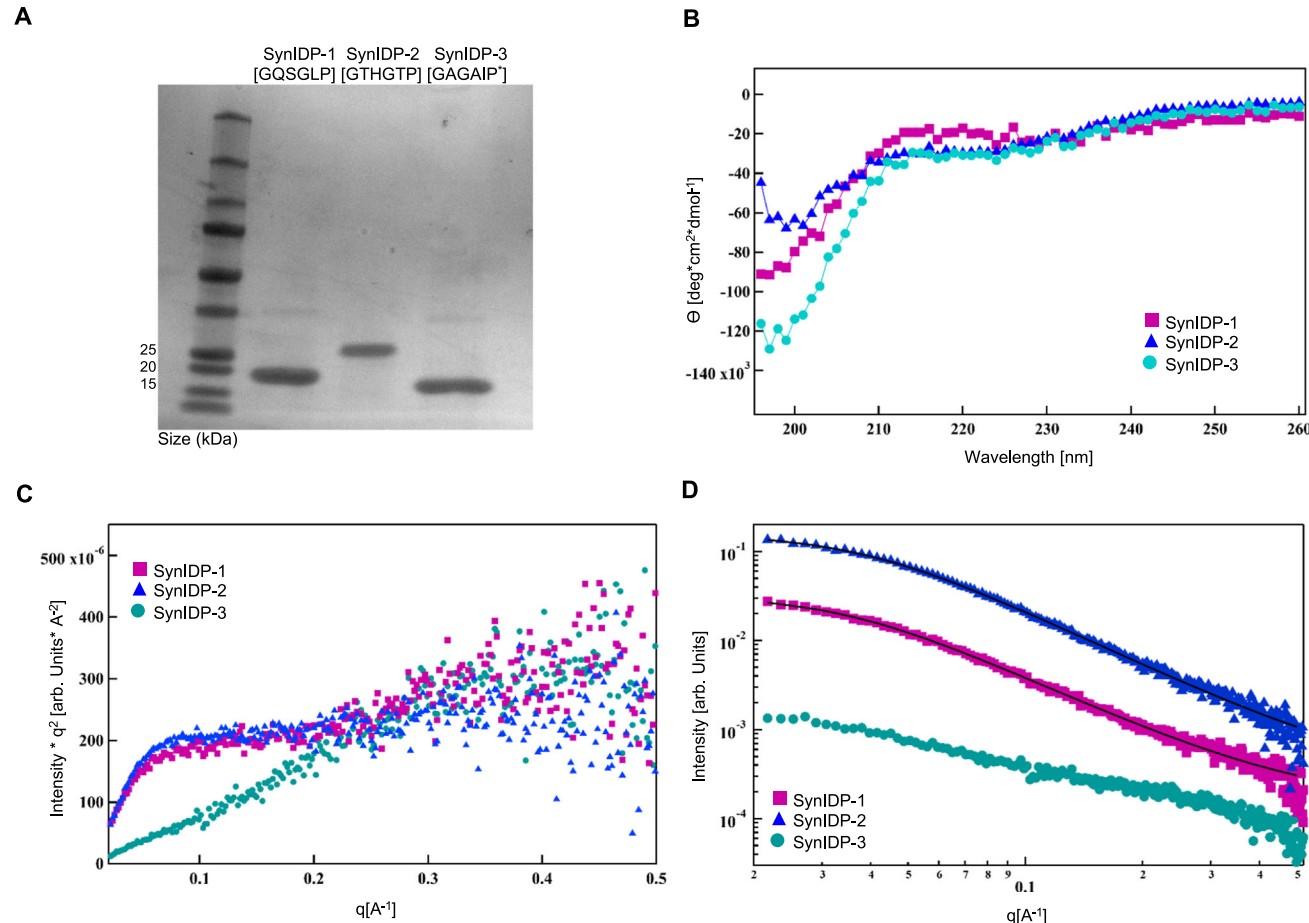

**Fig. 2 | Characterization of the physical and molecular properties of SynIDPs demonstrate that they are unstructured proteins. A** SDS-PAGE of purified SynIDPs. All SynIDPs run slightly higher than their expected molecular weight, which is characteristic of many IDPs. **B** CD spectra of the SynIDPs shows a random coil structure, which is deduced from the characteristic negative peak at 197 nm and a positive peak at 215 nm. **C** Kratky plots [I(q)·$q^2$ as a function of q] of the SynIDP reveal a characteristic of flexible disordered chain for SynIDP-1 and SynIDP-2 while demonstrating extended chain conformation for SynIDP-3. **D** Scattering profiles [I(q) vs. q] of SynIDPs in a log-log plot. Black lines represent Fit of the PEV model to the SAXS data of SynIDP-1 and SynIDP-2. Curves were offset along Y-axis for better visibility. Source data are provided as a Source Data files.

negative peak at ~197 nm and a positive peak at ~215 nm that are characteristic of a random coil[31,37]. As native IDPs are frequently known to have lower/upper critical solution temperature phase behavior (LCST/UCST) that causes their phase separation upon heating or cooling respectively[26,30,63], we carried out thermally ramped turbidity experiments in PBS and 1 M NaCl in a temperature range of 15–80 °C (Supplementary Fig. 12). No increase in absorbance was detected over this temperature range, indicating a lack of LCST phase separation under these conditions and ensures that no coacervation is expected due to elevated temperature or salt concentration.

Next, small angle X-ray scattering (SAXS) was performed to characterize the nanoscale structures of the SynIDPs. Displaying the scattering profiles of SynIDP-1 and SynIDP-2 as a Kratky plot [I(q)·$q^2$ as a function of q] (Fig. 2C) reveals a characteristic flexible disordered chain for both SynIDPs, as seen by the lack of a well-defined maximum and a plateau at large q values[64]. The Kratky plot of SynIDP-3 suggests an extended chain conformation, as seen by the upturn and uniform slope of the curve. This behavior can be explained by the missense mutation (G to E) every four repeats, which adds negative charges to the chain and restricts the available conformations to an extended chain due to intramolecular electrostatic repulsion. The scattering profiles of SynIDP-1 and SynIDP-2 were fit to a polymer excluded volume (PEV) model (Fig. 2D)[65–68], which describes the scattering from polymer chains and its mass fractal behavior. The PEV model was used to estimate the $R_g$ and fractal dimension of the SynIDPs, yielding $R_g$ of 4.2 ± 0.5 nm and 3.8 ± 0.3 nm for SynIDP-1 and SynIDP-2 accordingly. The obtained $R_g$ values are in good agreement with previously published data of $R_g$ for IDPs of a similar size[64]. The Porod exponent value for these two SynIDPs is 2, which indicates an idealized random walk chain (which can cross itself). The Porod exponent for SynIDP-3 was estimated from the slope in the mid-q range (0.03 < q < 0.2 A$^{-1}$) of the scattering profile (Fig. 2D) and was found to be 1, consistent with a rigid extended chain. As the PEV model assumes chain flexibility, it does not apply to the limit of a rigid extended chain (Porod exponent of 1) and could hence not be fit to SynIDP-3.

## Hydropathy of SynIDPs tags confirm their solubility

Previous attempts to rationally design SynIDPs for the purpose of solubility tags used the Wilkinson and Harrison solubility calculator[69,70] to predict the chance of soluble expression of a given protein in *E.coli*. For SynIDP-2, this calculator predicts 100% chance of solubility when overexpressed in *E.coli*. However, this calculator fails when calculating the solubility of SynIDP-1 and SynIDP-3, by predicting that there is 0% chance that these SynIDPs would be soluble.

Analysis of the data for expression of the SynIDPs fused to mTdT shows that the average residue hydropathy calculated using the Urry hydropathy scale[71] can be used to predict soluble vs. insoluble expression (Table 1). The Urry hydropathy scale delineates amino acids

**Table 1 | Summary of the solubility of SynIDPs and SynIDP-mTdT fusion proteins**

| SynIDP status | mTdT fusion status | Sequence | Urry scale Hydropathy |
|---|---|---|---|
| insoluble | | [GYGYAP]$_{24}$ | 12.4 |
| insoluble | | [GYGHAP]$_{28}$ | 27.3 |
| insoluble | | [GYVGTP]$_{24}$ | 28.8 |
| insoluble | | [GYMGKP]$_{24}$ | 33.6 |
| insoluble | | [GGMLAPGGMLAPGGMLAPGGMFAP]$_{6}$ | 35.5 |
| insoluble | | [GHIGVP]$_{24}$ | 36.9 |
| insoluble | | [GGMLAP]$_{24}$ | 37.5 |
| insoluble | | [GGLHTP]$_{28}$ | 41.0 |
| Soluble | | [GAGAIPGAGAIPGAGAIPVAGAIP]$_{6}$ | 42.3 |
| Soluble | insoluble | [GANMPQGASIPPGANIPPGASIPP]$_{6}$ | 43.2 |
| Soluble | | [GKFGTP]$_{24}$ | 44.6 |
| Soluble | | [GLSGSP]$_{28}$ | 45.6 |
| Soluble | | [GNGNVP]$_{24}$ | 45.8 |
| Soluble | insoluble | [GIGQAP]$_{28}$ | 46.3 |
| Soluble | soluble | [GQSGLP]$_{24}$ | 47.0 |
| Soluble | | [GIRGNPRLRGTPEYPGTPGIRGTP]$_{7}$ | 48.1 |
| Soluble | soluble | [GTHGTP]$_{24}$ | 49.2 |
| Soluble | soluble | [GAGAIPGAEAIPGAGAIPGAGAIP]$_{6}$ | 49.3 |

on a range from hydrophobic to hydrophilic based on the phase transition temperatures of a class of related SynIDPs−Elastin-like polypeptides (ELPs)−with the sequence (VPGXG)$_{n}$ where X is any residue except Pro and $n$ is the number of repeats. Given that the Urry scale is derived from measurements on ELPs−a SynIDP− rather than on folded globular proteins, we hypothesized that it would provide a better predictive tool to assess the hydropathy of the SynIDPs in this work. We used this scale to compute the hydropathy of the expressed SynIDPs selected by colony filtration in this study by averaging the scores of each amino acid. A ranking of hydropathies suggests that there is a threshold integer value of 42 that clearly separates insoluble (below 42) and soluble (above 42) SynIDPs. The threshold upon fusion to mTdT is shifted upwards to 47 (Table 1), suggesting that the greater hydrophobicity of mTdT relative to the SynIDPs requires its fusion to more hydrophilic SynIDPs to ensure soluble expression of the fusion protein.

**Fusion with SynIDPs enhances mTdT solubility and rescues enzymatic activity**

We repeated the expression of mTdT and the three soluble SynIDP-1/2/ 3-tev-H$_{6}$-mTdT constructs in *E. coli* BL21(DE3) by induction of protein expression with IPTG (0.5 mM) at an OD$_{600}$ of 0.7−1. To measure the amounts of mTdT in the insoluble and soluble fractions of the lysate, a Western blot using an anti-TdT antibody was performed (Fig. 3A). The Western blot image shows that mTdT is largely present in the insoluble fraction, with a small amount in the soluble fraction. In contrast, when mTdT is fused to SynIDP-1, SynIDP-2, or SynIDP-3, a significantly greater amount of the SynIDP-mTdT fusion is present in the soluble fraction. Next, the proteins were purified by IMAC on Ni-NTA gravity columns (Supplementary Fig. 13A), followed by further purification by

SEC (Supplementary Fig. 13B). We note that the efficiency of IMAC purification could be improved by moving the His$_{6}$ tag to the C-terminus of the fusion, which should improve its accessibility to the Ni-NTA ligand on the IMAC resin.

Murine-TdT is a eukaryotic enzyme that can promiscuously append nucleotides to the 3′-end of a single stranded DNA (ss-DNA) substrate[16]. We were interested in seeing whether the SynIDP-mTdT fusion retains the catalytic activity of mTdT. The fact that the proteins are soluble does not necessarily indicate that they are properly folded, as they could misfold by falling into a local minima−a trough−of free energy, which does not provide the correct tertiary structure to confer activity[72]. Additionally, fusion to a solubility tag could sterically interfere with the activity even if the enzyme were properly folded.

To explore the enzymatic activity of the recombinant mTdT and SynIDP-mTdT fusions, TdT-catalyzed enzymatic polymerization (TcEP) of ss-DNA was performed. Briefly, proteins were buffer exchanged into the optimal reaction buffer (50 mM Potassium Phosphate, 100 mM NaCl, 1 mM 2-mercaptoethanol, 0.1% Tween 20, 50% glycerol, pH 6.4) for nucleotide addition by mTdT to a final concentration of 0.5 or 1 μM (labeled as 1X or 2X) without cleaving off the SynIDP tag. A Cy5 labeled (dT)$_{50}$ oligonucleotide sequence was used as the primer to initiate the reaction with deoxythymidine triphosphate (dTTP) nucleotides at a 1:500 initiator: dTTP molar ratio[73]. The enzymatically catalyzed nucleotide addition reaction was allowed to proceed for 2 h at 37 °C, followed by heat inactivation of the enzyme at 95 °C prior to mixing with 2xTBE-Urea loading buffer to abolish any protein-DNA interactions. Reaction products were visualized by 10% TBE-Urea PAGE (Fig. 3C, Supplementary Fig. 13C). As a negative control, we used the initiator without any enzyme, while as a positive control we used commercially available TdT (Promega or NEB, working concentration

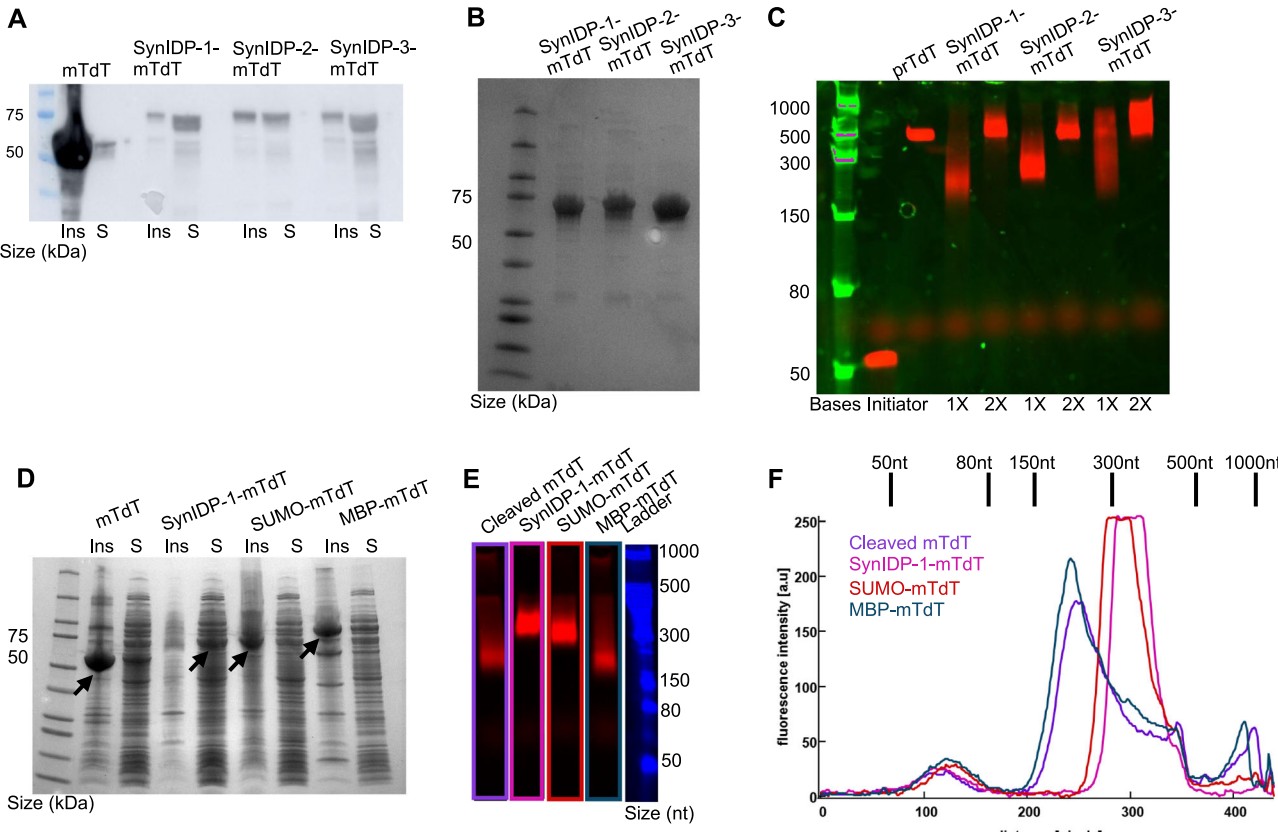

**Fig. 3 | Fusion to SynIDPs rescues soluble and functional expression of mTdT.**
**A** Western blot of insoluble and soluble fractions of mTdT and SynIDP-1/2/3-mTdT using anti-TdT antibody. **B** Purified SynIDP-1/2/3-mTdT after IMAC and SEC visualized on SDS-PAGE. **C** mTdT activity assay showing elongation of Cy5-poly-T$_{50}$ initiator on TBE-Urea PAGE gels. Low Range ssRNA ladder was used to quantify nucleotide addition. From left to right; ladder, initiator (negative control), Promega-TdT (positive control), SynIDP-1-mTdT 1X and 2X, SynIDP-2-mTdT 1X and 2X, SynIDP-3-mTdT 1X and 2X. **D** Insoluble (Ins) and soluble (S) fractions of mTdT, SynIDP-1-mTdT, SUMO-mTdT and MBP-mTdT visualized on SDS-PAGE. Large increase in soluble expression of mTdT are observed when fused to SynIDP-1 compared to fusion with SUMO or MBP. Arrows indicate the desired protein

product. **E** TdT activity assay showing elongation of Cy5-poly-T$_{50}$ initiator on TBE-Urea PAGE gels. Low Range ssRNA ladder was used to quantify nucleotide addition. From left to right; cleaved mTdT, SynIDP-1-mTdT, SUMO-mTdT, MBP-mTdT, Ladder. Image was processed from Supplementary Fig. 14D (see experimental). **F** ImageJ analysis of fluorescence intensity as measurement for dispersity and degree of polymerization of mTdT variants' reaction products. The peak at a distance of ~100 pixels stems from the dye front and is not a real product. SynIDP-1-mTdT outperforms the other variants by displaying lower dispersity and a higher degree of polymerization. The calibration of nt size to the distance on the gel (shown in the line above) was obtained by similar image analysis of the ladder. Source data are provided as a Source Data file.

of 1.2 U/µL). The recombinant mTdT that we purified from the soluble fraction demonstrated very little enzymatic activity (Supplementary Fig. 13C). In contrast, for all the SynIDP-mTdT fusions, the higher molecular weight DNA products are indicative of enzyme activity. Interestingly, when using 1 µM of the SynIDP-mTdT and a 1:500 initiator:dTTP ratio, we were able to achieve stoichiometric incorporation of ~500 nucleotides, similar to the length and polydispersity generated by the commercially available TdT (Promega). Importantly, we could do so without the need to cleave the SynIDP tags from mTdT.

To compare the performance of SynIDP with existing solubility tags, we expressed His$_6$-MBP-His$_6$-mTdT and His$_6$-SUMO-His$_6$-mTdT under the same conditions. Visualization of soluble and insoluble fractions of SynIDP-1-mTdT, SUMO-mTdT and MBP-mTdT (Fig. 3D) on SDS-PAGE demonstrates significant enrichment of the target protein in the soluble fraction of SynIDP-1-mTdT compared to soluble fractions of SUMO-mTdT and MBP-mTdT. SUMO-mTdT and MBP-mTdT also exhibited a larger fraction of insoluble mTdT expression than SynIDP-1-mTdT, highlighting the improvement in soluble expression of mTdT when fused to SynIDP-1 compared to SUMO and MBP. In addition to the constructs described above, a cleaved mTdT was generated via incubation of the soluble fractions of SynIDP-1-mTdT with SynIDP-1-TEV (further described in the following sections) at 4 °C overnight. The

constructs were purified using IMAC (Supplementary Fig. 14A, B) and SEC.

Enzymatic activity of the mTdT variants was measured by TcEP of ss-DNA (Fig. 3E, Supplementary Fig. 14D). To that end, all mTdT variants were buffer exchanged into the optimal reaction buffer (50 mM Potassium Phosphate, 100 mM NaCl, 1 mM 2-mercaptoethanol, 0.1% Tween 20, 50% glycerol, pH 6.4) to a final concentration of 0.5 µM (Supplementary Fig. 14C) or 1 µM (labeled as 1X or 2X). Recombinant, soluble mTdT demonstrated very little activity (Supplementary Fig. 14D) as mentioned above. In contrast, all of the mTdT variants expressed as fusions to solubility tags, as well as the cleaved mTdT from SynIDP-1-mTdT demonstrated a marked improvement in nucleotide addition compared to recombinant mTdT. Further analysis of the TcEP products was performed by image processing of the original gel (Supplementary Fig. 14D, Fig. 3E) to obtain the fluorescence intensity distribution for each of the mTdT variants (Fig. 3F) as function of distance from the gel bottom. For the TcEP reaction, we were interested in obtaining a high degree of polymerization, which is indicated by a higher distance from the bottom of the gel, and low dispersity, which can be evaluated by the width of the fluorescence signal. Comparing the fluorescence intensity curves of the mTdT variants shows that SynIDP-mTdT outperforms the other mTdT variants; SynIDP1-mTdT shows the highest degree of polymerization and the

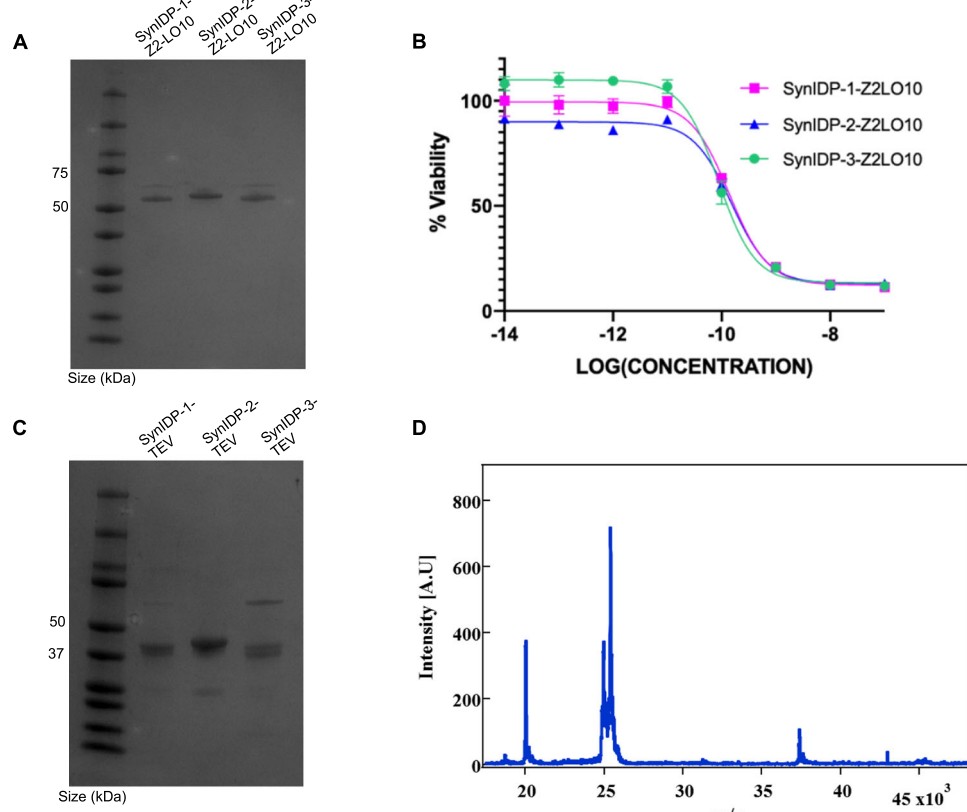

**Fig. 4 | Fusion to SynIDPs rescues soluble and functional expression of Z2-LO10 and TEV proteins. A** Purified SynIDP-1/2/3-Z2-LO10 after IMAC and SEC. **B** Log-fold dilutions of SynIDP-Z2-LO10 and SynIDP controls (see Supplementary Fig. 15C) were incubated with CT-2A-EGFRviii, an EGFR positive murine glioma cell line for 48 h and tested for viability by an MTS assay. $n = 3$ replicates, error bars represent SD. **C** Purified SynIDP-1/2/3-TEV after His-purification and SEC. **D** MALDI-TOF MS of products of SynIDP-2-TEV reaction with ELP-tev-FGF21 substrate at 37 °C for 30 min confirms proteolytic activity. The absence of the intact substrate is evident by the lack of a peak at m/z 45,459. The cleaved FGF-21 (m/z of 20,038) and the ELP-tev (m/z of 25,439) can be seen in the MS spectra. The peak at m/z ~ 37,000 matches the expected Mw of SynIDP-2-TEV. Mass spectra for the control and products of SynIDP-1/3-TEV reaction with ELP-tev-FGF21 are shown in SI (see Supplementary Fig. 18). Source data are provided as a Source Data file.

lowest dispersity (Fig. 3E, F), indicating the superior folding of mTdT within the cohort of solubility tags tested.

### Fusion with SynIDPs enhances LO10 solubility and rescues exotoxin activity

We next tested the ability of SynIDP-1/2/3 tags to promote soluble protein expression with a second inclusion body forming protein. The LO10 domain derived from *P. aeruginosa* exotoxin A is a powerful ribosome inhibitor that leads to cellular death upon internalization, and its expression alone results in inclusion body formation[19,74]. We fused the LO10 exotoxin A domain to the C terminus of a dimeric EGFR targeting affibody domain (Z2), as the free C terminus of the exotoxin is crucial for cytotoxic activity, with the intent that the Z2-LO10 construct could target and kill tumor cells that overexpress EGFR such as certain breast cancers and gliomas[75,76].

Fusion of SynIDP-1/2/3 to Z2-LO10 led to a marked improvement in the soluble expression of the protein at 37 °C compared to the protein without the SynIDP tags (Supplementary Fig. 15A). Soluble proteins were purified by IMAC (Supplementary Fig. 15B) followed by SEC, concentrated and buffer exchanged into PBS (Fig. 4A) prior to testing in vitro activity in an EGFR-expressing glioma line, CT2A-EGFRviii. Incubation of tumor cells with SynIDP-Z2-LO10 shows concentration dependent cytotoxicity (Fig. 4B), with IC50's ranging from 75 to 125 pM, whereas treatment with the SynIDP tags alone caused no cell death (Supplementary Fig. 15C). These data indicate that all three SynIDP fusions exhibit potent cytotoxicity that is imparted by the exotoxin domain. These data hence clearly show that fusion of Z2-LO10 to these SynIDPs enables soluble and functionally active expression of the protein.

### Fusion with SynIDPs enhances TEV protease solubility and rescues proteolytic activity

Finally, we tested the three SynIDPs with TEV protease, a protein that also forms inclusion bodies when overexpressed in *E. coli*[11,21,62]. TEV protease is used as a reagent to cleave off target proteins of interest from their fusion partners such as solubility or purification tags[77]. We have previously synthesized and validated a TEV substrate comprising of an ELP, a tev cleavage site, and fibroblast growth factor-21 (ELP-tev-FGF21), which was used to quantify the activity of the SynIDP-TEV fusions[78].

Fusion of SynIDP-1, SynIDP-2 and SynIDP-3 to TEV protease with a terminal His6 tag and no tev cleavage site between SynIDP and TEV protease led to a marked improvement in the solubility of TEV when expressed at 37 °C compared to the protein alone (Supplementary Fig. 16A, Supplementary Fig. 8A). Soluble SynIDP-TEV fusions were purified from bacterial lysate by IMAC and SEC (Supplementary Fig. 16B), prior to concentration and buffer exchange into 50 mM Tris, 0.5 M EDTA, pH 8.0 buffer (Fig. 4C). A 1:100 molar ratio of enzyme to the substrate—ELP-tev-FGF21 (Supplementary Fig. 17)—was incubated at 37 °C for 30 min prior to heating to 95 °C for 2 min to inactivate the enzyme. Cleaved fragments were identified by MALDI-TOF MS (Fig. 4D, Supplementary Fig. 18), which showed a peak at m/z ~ 20000, which was not present in the control (ELP-tev-FGF21 at same conditions with no addition of SynIDP-TEV), indicative of the cleaved FGF21 as well as

disappearance of the peak at m/z ~ 45000 indicative of the absence of full length ELP-tev-FGF21 protein.

## Discussion

We have developed a pooled library approach that uses RCA to create large and complex gene libraries of repetitive motifs from libraries of microarray synthesized oligonucleotides. Although oligonucleotides from DNA microarrays have been used as starting material for overlap-based gene synthesis of nonrepetitive double stranded DNA fragments, we have demonstrated the use of microarray oligonucleotides and RCA to generate controllable sized DNA products that are further used to express a heterogenous library of synthetic IDPs. Utilizing microarray synthesized oligonucleotide pools allowed for one-pot synthesis of thousands of genes, which can significantly reduce the cost of DNA library synthesis compared to traditional solid-phase synthesis. The primary cost consists of the initial purchase of oligonucleotides (~$1k per library) and DNA sequencing. We estimated the total cost for 500 genes via the pooled library strategy to be ~$4k, compared to the estimated cost of commercial DNA synthesis of ~$25k. Furthermore, this unique RCA based approach for the cloning purposes of SynIDPs proved useful to evolve beneficial mutations that are otherwise hard to incorporate[79]. An example of this is SynIDP-3 ([GAGAIPGAEAIPGAGAIPGAGAIP]$_6$), which contains a G to E mutation every 6 repeats and is considerably more soluble than [GAGAIP]$_{24}$. The a priori identification of these mutations and their incorporation into an otherwise highly repetitive gene is almost impossible to achieve by other methods for the assembly of repetitive genes, many of which we have previously developed[38,80–82].

Starting from a DNA library size of 1020 unique gene sequences verified by NGS, we screened for cloning artifacts and soluble expression, and identified SynIDPs that showed high levels of soluble expression in *E. coli*. These SynIDPs were designed to have high proline and glycine content to ensure a lack of secondary structure and achieve high solvation, driven by a combination of proline's fixed φ dihedral angle and glycine's conformational entropy[83–86]. The lack of secondary structure and the high conformational flexibility of the SynIDPs was validated using CD and SAXS, with SynIDP-3 having a more rigid structure as compared to SynIDP-1/2 due to a G to E mutation every 6 repeats, which increases the negative charge and hence repulsion along the chain.

SynIDP-1/3 are good examples of the superiority of libraries and experimental screening for solubility over a rational design approach, because SynIDP-1/3 could not have been rationally designed for this purpose, as according to Wilkinson and Harrison solubility calculator[69,70] they would be expected to be insoluble. As an alternative calculator for predicting SynIDPs solubility, we suggest using Urry hydropathy scale[71]. The most soluble synthetic IDPs that have been experimentally identified in this work, do not contain aromatic residues. This outcome can be expected, given that aromatic residues are well known to promote aggregation due to π stacking and hydrophobic interactions[87]. Counterintuitively, the most soluble synthetic IDPs also lack positive charge. We speculate that SynIDPs that have high positive net charge may complex with other negatively charged macromolecules such as DNA and precipitate out even though they have a high hydropathy (low hydrophobicity). Similarly, Chan et al. demonstrated that soluble expression of proteins correlates with a lack of positively-charged surface[88].

To demonstrate the utility of these relatively small ~15 kDa and highly soluble IDPs, three different SynIDPs were fused to a set of known inclusion body forming proteins to promote their soluble overexpression in *E. coli*. We note that this is an emerging and interesting application of IDPs, as there is little previously published work on native IDPs as tags to promote soluble protein expression[24,25,89]. However, native IDPs are not designed—or optimized—by nature to serve as fusion tags and their native function can be toxic to cells, if overexpressed[90].

We demonstrate that active mTdT with high enzymatic processivity can be solubly expressed and purified using SynIDP as a fusion tag, and more importantly, that the SynIDP-mTdT fusion retains the activity of mTdT without the need to cleave the tag, which greatly lowers the cost and improves the efficiency of its production. When the tag is cleaved, mTdT demonstrates improved enzymatic activity over its recombinant counterpart, indicating that SynIDPs promote correct folding of the fusion partner. This finding highlights the utility of SynIDPs as solubility tags, as they are small, unstructured proteins that minimally interfere with biological activity. Additionally, when compared to common tags such as MBP or SUMO, our SynIDP outperformed both tags in improving soluble expression of mTdT and retaining activity of the enzyme. We also screened these SynIDPs with two other inclusion body forming proteins— Z2-LO10 and TEV protease—to demonstrate the utility of these tags to promote the soluble expression of these inclusion body forming proteins. Overall, the three proteins span a range of molecular weights from 25 to 59 kDa and isoelectric points from 5.2 to 8, indicating they may be useful for enhancing the soluble expression of other recombinant proteins as well.

As a potential future direction, the SynIDPs and the technologies developed here may be useful for the soluble expression of cysteine-rich proteins (otherwise forming inclusion bodies) in *E. coli* cell lines that enable disulfide formation, such as SHuffle cells[91]. Another application of these highly soluble IDPs is to potentially extend the biological half-life of protein-based therapeutics that they are fused to[92,93]. In conclusion, this work is notable because it demonstrates a method for the cost-effective one-pot, multiplexed synthesis of a library of genes of repetitive polypeptides and applies this technology to identify SynIDPs that promote the soluble expression of proteins that are otherwise known to form inclusion bodies in *E. coli*.

## Methods

### Gene library synthesis using RCA

A tube of OligoMix® was ordered from LC Sciences, containing approximately 3.5 pmol of 1020 unique specified 72-nt sequences. A 10 µL T4 polynucleotide kinase (PNK) reaction was prepared containing 1 pmol 72-nt oligonucleotide mix, 1 X T4 Ligase buffer, and 10 U T4 PNK. The reaction was incubated at 37 °C for 30 min and inactivated at 65 °C for 20 min. A 20 µL CircLigase II reaction was prepared containing the entire 10 µL PNK-treated oligonucleotide mix, 200 U CircLigase II, 1X CircLigase II Buffer, and 2.5 mM MnCl$_2$. The reaction was incubated at 60 °C for 2 h and inactivated at 80 °C for 10 min. To remove linear ssDNA, a 12 µL exonuclease reaction was prepared containing 10 µL CircLigase II product, 20 U Exonuclease I, and 200 U Exonuclease III. The reaction was incubated at 37 °C for 1 h and inactivated at 80 °C for 15 min. A 36 µL annealing reaction was prepared containing 2 nmol Exo-resistant random primer (Thermo Fisher Scientific), 1.1X RephiPHI buffer, and 2 µL Exonuclease reaction product. The annealing reaction was incubated in a thermal cycler with a slow temperature ramp from 95 °C to 4 °C, holding for 1 min every at every 2.3 °C increment. A 40 µL RCA reaction was prepared containing 36 µL annealing product, 50 nmol dNTPs mixture where 25% of the dCTP was replaced with methyl-dCTP, 200 U RepliPHI polymerase. The RCA reaction was incubated at 30 °C for 4 h and inactivated at 65 °C for 20 min. The resulting DNA was then purified via isopropanol precipitation. A 25 µL digestion reaction was prepared containing 5 U SexAI, 2 µL of purified DNA, and 1X NEB Cutsmart buffer. The reaction was incubated overnight at 37 °C. The reaction was visualized on a 1% low melting point agarose gel and a 400−600 bp band was gel-purified using the QIAquick gel extraction kit (Qiagen).

A SexAI recognition site (ACCTGGT), and DNA that encode a N-terminal leader sequence encoding MSKGP to optimize expression,

a TEV protease cleavage site, and a His$_6$-tag were added to a modified pET-24a(+) vector using PCR. The construction of this modified vector from pET-24a(+) (Novagen) has been previously documented[80]. The amino acid sequence encoded by the gene after ligation is: MSKGP-[PGXXXX]$_n$-GENLYFQGHHHHHHG. 2.5 ng of the plasmid was linearized and amplified in a 50 μL PCR reaction containing 250 μM of each dNTP, 1X Herculase II PCR buffer, 0.25 μM each of universal forward (ATATATATACCAGGTGAAAACCTGTATTTTCAGGGCCATCACCATCA CCATCACGGCTAA) and reverse (ATATATATACCTGGTCCTTTGCTCA TATGTACTCC) primers and 0.5 μL Herculase II Fusion polymerase. The reaction was incubated at 95 °C for 2 min, followed by 5 cycles at 95 °C for 20 s, 56 °C for 20 s and 72 °C for 2 min 40 s, followed by 25 cycles at 95 °C for 20 s and 72 °C for 3 min, and a final step at 72 °C for 3 min. The linearized vector was gel-purified from a 1% low melting point agarose gel using the QIAquick gel extraction kit (Qiagen). A 90 μL digestion reaction was prepared containing 50 U SexAI, 5 μg of purified DNA, 3 U shrimp alkaline phosphatase (rSAP), and 1X Cutsmart buffer. The reaction was incubated overnight at 37 °C. The resulting reaction was electro-eluted from a 1% low melting point agarose gel using a BioRad Model 422 Electro-Eluter.

A 10 μL ElectroLigase reaction was prepared containing 1 μL ElectroLigase, 1 X ElectroLigase reaction buffer, 20 ng -6 kbp linearized vector and 10 ng of -0.5 kbp library insert. The ElectroLigase reaction was incubated at room temperature (25 °C) for 30 min and inactivated at 65 °C for 15 min. The resulting mixture was diluted 1:3 with distilled water and 1 μL was transformed into 20 μL ElectroMAX DH10B electrocompetent *E. coli* cells (≥10$^{10}$ cfu/μg) using a BioRad MicroPulser Electroporator.

In total, 4 × 100 μL of SOC outgrowth medium was spread on four 100 mm diameter LB-kanamycin agar plates. Following overnight growth at room temperature, the resulting colonies from each plate were covered with 1 mL of LB-kanamycin and gently scraped off with cell platers. The cell suspensions were removed from the plates with a pipette and combined in a single 15 mL conical tube. Plasmids were isolated from the cell suspensions by a QIAprep spin miniprep kit (Qiagen).

### NGS analysis of gene library
Illumina MiSeq was used to analyze the sequence content of the gene library prior to screening. Illumina overhang adapters were added by PCR using region-specific primers surrounding the coding region. 12.5 ng of plasmid DNA was linearized and amplified in a 25 μL reaction containing 0.2 μM of each of the forward and reverse primers and 1 X KAPA Hifi HotStart ReadyMix. The reaction was incubated at 95 °C for 3 min, followed by 8 cycles at 98 °C for 30 s, 55 °C for 30 s and 72 °C for 30 s, and a final step at 72 °C for 5 min. 6 reactions were pooled together and Ampure XP bead cleanup was used to remove fragments greater than 500 bp. Next, indices and Illumina sequence adapters were added using the Nextera XT Index Kit using the same PCR reaction conditions as above. During Miseq sequencing, 20% PhiX genome was added to counterbalance the relatively low complexity and diversity of the gene libraries compared to genome libraries.

### Colony screening for solubility and expression (CoFi)
One ng of miniprep plasmid was transformed into 20 μL Acella™ electrocompetent BL21(DE3) *E. coli* cells using the BioRad Micropulser Electroporator. The SOC outgrowth medium was divided into 10 LB-kanamycin plates per transformation, resulting in about 500 colonies per plate after overnight growth at 37 °C. The plates were replica plated twice and grown overnight at 37 °C. CoFi blotting was performed on one set of replica plates (Supplementary Fig. 3)[42]. Colonies were transferred to a Durapore membrane. The Durapore membrane was transferred, colonies-side up, to an induction plate containing kanamycin and 200 μM IPTG for 4 h at 37 °C. The Durapore membrane was then transferred over a filter sandwich consisting of a nitrocellulose membrane and a Whatman 3MM paper drenched in B-PER lysis buffer containing 5 U/mL DNase I and 100 μg/mL lysozyme. The filter sandwich was incubated for 30 min at room temperature. The blot was then freeze-thawed three times at −80 °C for 10 min and then at 37 °C for 10 min. The nitrocellulose membrane was then subject to antibody detection via standard Western blot analysis protocol, using a 1:4000 dilution of DyLight 650 conjugated 6x-His epitope tag antibody (Life Technologies, cat. no. MA1-21315-D650). The resulting blot was imaged using a Typhoon 9410 scanner (GE Healthcare).

Image thresholding was performed in ImageJ to identify colonies containing soluble protein. The resulting binary image was printed, and the second replica plates were placed over the printed image and aligned. The binary image was then used to locate and pick colonies. Picked colonies were grown in 96 deep-well polypropylene microplates overnight at 37 °C and 200 rpm. Clones were screened for frameshifts, empty vectors, and duplicates with direct sequencing from colonies using T7 sequencing primers. Unique clones were saved for protein purification.

### Determining solubility and expression of SynIDPs by dot blot and PAGE
The Duetz Microflask system was used to grow screened clones in 24 deep-well microplates holding 14 mL round-bottom tubes containing 2 mL of Overnight Express TB (EMD Millipore) plus 100 μg mL$^{-1}$ kanamycin[94]. A clamp system and sandwich covers were used to allow for high-frequency orbital shaking and high oxygen transfer rates of 1–2 culture volumes per minute without well-to-well contamination. After overnight growth at 30 °C and 1200 rpm on a VWR incubating mini shaker, chemical lysis was performed by resuspending cell pellet in 900 μL of B-PER containing 3.6 U DNase I and 90 μg lysozyme at room temperature. Three freeze-thaw cycles were performed at −80 °C for 15 min and room temperature for 15 min.

For screening of cell lysates, 30 μg of total cell lysates were separated by centrifugation. The insoluble pellet was resuspended in 100 μL of 8 M urea. 10 μL of the soluble fraction and 10 μL of the insoluble fraction were subject to a dot blot on a nitrocellulose membrane, using the Bio-Dot® Microfiltration Apparatus (Bio-Rad Laboratories). The membrane was then subject to His$_6$-tag antibody detection via standard Western blot analysis protocol. The blot was imaged using a Typhoon 9410 scanner (GE Healthcare), and ImageJ was used to quantify the average intensities. For PAGE analysis, 10 μL of the soluble fraction and 10 μL of the insoluble fraction were separated by SDS-PAGE on a TGX precast gel and then stained with EZBlue gel staining reagent (Sigma). The resulting gel was imaged with a Gel Doc XR system.

### Gibson assembly of SynIDP-mTdT DNA
Universal forward (GTCCAACATCAATACAACCTATTAATTTCCC) and reverse (TGCACAGCTTGCAGCGGGTCGCCGTGGTGGTGATGATGAT GGCCCTGAAAATACAGGTTT) primers were designed to amplify approximately half of the SynIDP-tev-H$_6$-encoding plasmids, serving to form the N-terminal coding sequence (CDS) of the fusion constructs and an upstream vector region. Additionally, an extra overhang sequence was incorporated in the reverse primer with a mTdT-overlapping sequence for Gibson assembly. Plasmids containing each gene were amplified with their respective primers in 50 μL PCR reactions, which contained 0.25 ng of plasmid template, 250 μM of each dNTP, 1X Herculase II PCR buffer, 0.25 μM each of universal forward and reverse primers and 0.5 μL Herculase II Fusion polymerase. The reactions were incubated at 98 °C for 3 min, followed by 30 cycles of 98 °C for 20 s, 60 °C for 20 s and 72 °C for 30–60 s, and a final step at 72 °C for 3 min. The DNA sequence that encodes murine TdT was commercially synthesized and inserted (Integrated DNA Technologies) into the pET-24a(+) vector (Novagen)[54]. Forward

(GGTGAAAACCTGTATTTTCAGGGCCATCATCATCACCACCACGGC-GAC) and reverse (CACTTGATAACCTTATTTTTGACGAGGGGAAATTAATAGGTTG) primers were designed to amplify approximately half of the mTdT-encoding plasmid, serving to form the C-terminal coding sequence (CDS) of the fusion constructs and a downstream vector region which partially overlaps the previous amplicon and fills the remaining omitted vector sequence. Additionally, an extra overhang sequence was incorporated in the forward primer with a tev-$H_6$-overlapping sequence for Gibson assembly. All PCR products were then purified with the AxyPrep Mag PCR clean-up kit (Axygen) and quantified on a NanoDrop 1000 spectrophotometer (Thermo Scientific) by the absorbance at 260 nm. Both amplicon fragments at about 0.02 pmol each were combined in a 10 μL Gibson assembly reaction containing 1X Gibson assembly master mix (New England Biolabs). The reaction was incubated at 50 °C for 1 h. 2 μL of a 1:4 dilution of each Gibson product was transformed into 50 μL EB5α competent cells following the manufacturer's instructions. The expected amino acid sequence of the final gene after cloning was: MSKGP-[GXXXXP]$_n$-GENLYFQGHHHHHHG-mTdT. Successful clones were screened with by fluorescent Sanger sequencing. Plasmids were then isolated from sequence-verified colonies using a QIAprep spin miniprep kit.

### Construction and assembly of SynIDP-Z2-LO10 DNA
A similar PCR strategy was used as above to generate a linearized pET-24a(+) vector that contained matching overhangs to a commercially synthesized Z2-LO10 gene (Supplementary Fig. 18) (IDT). Gibson assembly was performed, as described above, and the Gibson product was transformed into 50 μL EB5α competent cells following the manufacturer's instructions. The expected amino acid sequence of the final gene after cloning was: MSKGP-[GXXXXP]$_n$-GENLYFQGHHHHHHG-Z2-LO10. Successful clones were identified by fluorescent Sanger sequencing and plasmid DNA were then isolated from sequence-verified colonies using a QIAprep spin miniprep kit.

### Construction and assembly of SynIDP-TEV DNA
SynIDPs were generated by PRe-RDL to reconstruct SynIDPs without the tev cleavage site[80]. TEV protease was appended onto the C terminus of the SynIDPs by Pre-RDL such that the expected amino acid sequence of the final gene was: MSKGP-[GXXXXP]$_n$-GHHHHHHG-TEV Protease. Successful clones were identified by sequencing and plasmids were isolated from sequence-verified colonies using a QIAprep spin miniprep kit.

### Construction and assembly of MBP-mTdT and SUMO-mTdT genes
Genes from MBP and SUMO were identified from UniProt (Accession: P0AEX9 and B7G0R2, respectively) and codon optimized for *E.* coli expression with a His$_6$ tag at the N-terminus. The resulting genes were cloned into a pET-24a vector via Gibson Assembly and subsequently ligated with a His$_6$-mTdT via Pre-RDL to generate His$_6$MBP-His$_6$mTdT or His$_6$SUMO-His$_6$mTdT.

### Protein expression and purification
**Expression and purification of SynIDPs.** Following plasmid isolation of sequence-verified clones, plasmids were transformed into BL21(DE3) competent *E. coli* cells (NEB). Glycerol stocks of BL21(DE3) *E. coli* harboring the desired genes supplemented with kanamycin were kept at -80 °C. Starter cultures (4 ml 2xYT medium, 45 μg/ml kanamycin) were inoculated from these stocks, grown overnight and used to inoculate 1 L of 2xYT medium containing kanamycin (45 μg/ml). The cultures were shaken (200 rpm) at 37 °C to an OD$_{600}$ of 0.8 and induced by the addition of 0.5 mM IPTG and allowed to grow overnight in the shaker incubator. Cells were harvested by centrifugation (5k rcf, 15 min, 4 °C) and resuspended in 25 mL of 100 mM PBS buffer (pH 7.6). *E. coli* cells were lysed using a tip sonicator (Fisher Scientific) following

by centrifugation (18k rcf, 25 min, 4 °C). Supernatants (termed soluble fraction) were subjected to purification.

**IMAC purification of SynIDPs.** 3 ml of HisPur Ni-NTA resin was added to a propylene gravity column. 30 mL of 100 mM PBS buffer (pH 7.6) were added to the column, followed by 3 rounds of addition of the total supernatant volume (25 mL) to the column (the flow through was collected and added back to the column). After the third time the flow through was collected (termed ft) the column was washed with 30 mL of PBS containing 25 mM imidazole (collected, termed W1), followed by washing with 10 mL PBS containing 50 mM imidazole (collected, termed W2). The elution step was performed with 5 ml PBS containing 100 mM imidazole (collected, termed E). All fractions were visualized on SDS protein gel, and the relevant fractions were pulled for next step SEC purification.

**Size exclusion chromatography and dialysis of SynIDPs.** IMAC purified SynIDPs were further subjected to SEC on a HiLoad 26/600 Superdex 200 pg column at 2.0 mL/min and 4 °C. Protein fractions were tracked through the absorbance at 220 nm and 280 nm, and fractions were collected in 3 mL aliquots. Protein fractions were run on a 4-20% TGX protein gel to confirm the desired product, and fractions containing the desired product were pooled together. The pooled fractions were then transferred into dialysis tubing (3.5 MWCO, Thermo Fisher) and dialyzed against DI water followed by lyophilization of the SynIDP and storage at RT until further use.

**Expression and purification of SynIDP-Z2-LO10 and SynIDP-TEV.** The expression of both SynIDP-Z2-LO10 and SynIDP-TEV was similar to the process described above, except that harvesting of cells occurred 4 h after induction instead of overnight. SynIDP-TEV was purified in PBS supplemented with 1 mM β-mercaptoethanol. SynIDP-Z2-LO10 was harvested and purified in PBS.

**Expression and purification of mTdT, SynIDP-mTdT, MBP-mTdT, and SUMO-mTdT.** The expression of mTdT, SynIDP-mTdT, MBP-mTdT, and SUMO-mTdT was similar to the process described above with the following changes: the 1 L cultures were shaken (200 rpm) at 25 °C to an OD$_{600}$ of 1.0 and induced by the addition of 0.5 mM IPTG. Upon induction, the temperature was immediately reduced to 16 °C and the cultures were incubated overnight in the shaker incubator. Cells were harvested by centrifugation (5k rcf, 15 min, 4 °C) and resuspended in 25 mL ice-cold buffer (50 mM potassium phosphate (pH 7.6), 100 mM NaCl 1 mM β-mercaptoethanol, 0.1% Tween20 and 5 mM imidazole)− named the "Lys buffer". *E. coli* cells were lysed by sonication (Fisher Scientific) and insoluble debris was separated by centrifugation (18k rcf, 25 min, 4 °C). The pellets were resolubilized in 25 mL 2% SDS (termed insoluble fraction). Supernatants (termed soluble fraction) were subjected to purification.

**IMAC purification of mTdT, SynIDP-mTdT, MBP-mTdT, and SUMO-mTdT.** Similar to the description above, while replacing PBS buffer at the entire process with ice-cold buffer (50 mM potassium phosphate (pH 7.6), 100 mM NaCl 1 mM β-mercaptoethanol, 0.1% Tween20) and performing the process at 4 °C.

**SEC and buffer exchange of mTdT, SynIDP-mTdT, MBP-mTdT, and SUMO-mTdT.** SEC was performed similarly to SEC of SynIDPs described above, while replacing PBS with ice-cold buffer (50 mM) potassium phosphate (pH 7.6), 100 mM NaCl 1 mM β-mercaptoethanol, 0.1% Tween20. The pooled fractions were then concentrated using Amicon Ultra-15 centrifugal filters (30 kDa MWCO, Sigma) and were buffer exchanged into the desired storage and activity buffer (100 mM potassium phosphate (pH 6.4), 200 mM NaCl, 2 mM β-mercaptoethanol, 0.2% Tween20).

**Western blotting**. Equal volumes of soluble and insoluble fractions were diluted in PBS, heated to 98 °C for 1 min and then separated by SDS-PAGE on Mini-PROTEAN TGX precast gels. The presence of the mTdT was determined by Western blot analysis, using a 1:2000 dilution of anti-TdT mouse monoclonal antibody (clone: 7H9A33, BioLegend, VWR, cat. no. 10763-336) as the primary antibody and 1:6,667 dilution of HRP conjugated IgG (H + L) Goat anti-Mouse (Invitrogen 31430). The signal was developed using Thermo Scientific™ SuperSignal™ West Pico PLUS chemiluminescent substrate. The resulting Western blot was imaged on an Amersham imager 600 (GE Healthcare).

### Circular dichroism

The secondary structure of the SynIDPs was characterized by CD spectroscopy, which was performed using an Aviv Model 202 instrument and a 1 mm quartz sample cells (Hellma). Purified proteins were diluted to a final concentration of 50 µM in 10 mM PBS, pH 7.4. The CD spectra were obtained at 37 °C from 260 nm to 190 nm in 1 nm steps with a 2 s averaging time. The CD spectra were corrected for the 10 mM PBS buffer signal at 37 °C and average was calculated based on 3 runs.

### Mass spectrometry

Samples were analyzed using Bruker Autoflex LRF matrix-assisted laser desorption ionization tandem time of flight mass spectrometer (MALDI-TOF MS). 2 µL of sample was mixed with 2 µL of a saturated solution of α-Cyano-4-hydroxycinnamic acid (HCCA) matrix and deposited on a MALDI plate. At least 3 different expression batches were analyzed for each protein. Myoglobin was used as reference.

### Small angle X-ray scattering (SAXS)

SAXS patterns of SynIDPs (1 wt.%) were obtained with a SAXSLAB GANESHA 300-XL at a range of $0.012 < q < 0.6\,\text{Å}^{-1}$. CuKα radiation was generated by a Genix 3D Cu source with an integrated monochromator, three-pinhole collimation, and a 2D Pilatus 300 K detector. Solutions were placed into a 1.0 mm quartz glass capillary (wall thickness of 0.01 mm). Measurements were performed under vacuum at ambient temperature. The scattering curves were corrected for counting time and sample absorption. The scattering spectra of the solvent was subtracted from the corresponding solution data using the Irena package for analysis of small-angle scattering data[95]. Data analysis was based on fitting the scattering curve to an appropriate model by SasView program.

### mTdT cleavage assay

SynIDP-1-mTdT and SynIDP-1-TEV were expressed as described above. The soluble fractions of the two proteins were mixed in a 1:1 v:v ratio and incubated at 4 °C overnight. The resulting mixture was purified via IMAC, as described above, and visualized by SDS-PAGE. The elution fraction harboring cleaved mTdT was further purified by SEC, as described above, to separate the desired product from other contaminants.

### TdT activity assay

To study the activity of mTdT, cleaved mTdT, TdT-fused IDPs, MBP-mTdT, and SUMO-mTdT, TdT-catalyzed enzymatic polymerization (TcEP) was performed. A 10 µL polymerization reaction was prepared containing 0.5 µM (1X) and 1 µM (2X) final concentrations of each of the purified mTdT variants, 250 µM of the monomer-deoxythymidine triphosphate (dTTP), 0.5 µM of initiator- Cy5-$(dT)_{50}$, and 1X TdT reaction buffer (Promega). The reaction product was heat inactivated at 95 °C for 2 min prior to mixing with 2x TBE-Urea loading buffer to abolish any protein-DNA interactions. Cy5 labeled poly(dT) was then visualized on 10% TBE-UREA PAGE using an Amersham imager 600 (GE Healthcare) with Cy5 filters[56]. Low

range ss-RNA ladder (NEB) was used to quantify nucleotide additions by all mTdT variants. Promega TdT and NEB TdT were used in a working concentration of 1.2 U/µL.

Image J was used for image analysis, as follows. First, the gel image was split into 3 RGB channels and each lane was duplicated and saved as a separate image followed by creating a profile plot. For the mTdT variants the red channel was used for image analysis, whereas for the Mw ladder the blue channel was used.

### Z2-LO10 functional assay

Murine glioma CT-2A-EGFRvIII (a gift from the Bigner lab at Duke University) were cultured in DMEM-High Glucose supplemented with 10% FBS, Hydromycin, and Geneticin according to ATCC passaging instructions. Cells were seeded at 50,000 cells/well, and serial log dilutions of purified IDP-Z2-LO10 ranging from $10^{-7}$ to $10^{-14}$ M were added to wells prior to incubation for 48 h. 10% v/v of CellTiter 96 Aqueous one solution (Promega) was added to each well and plates were read over 3 h using a UV-Vis spectrophotometer (Ticon) at 650 nm. Experiments were performed in triplicates.

### TEV protease functional assay

Prior to each activity assay, we buffer exchanged SynIDP-TEV in PBS to a 50 mM Tris, 0.5 M EDTA, pH 8.0 buffer. An ELP fused to FGF21 with a TEV cleavage site (named tev) between the two moieties was used as the substrate to demonstrate activity of the purified SynIDP-TEV. The SynIDP-TEV was incubated with substrate at 1:100 molar ratio at 37 °C for 30 min, prior to heat shocking at 95 °C for 2 min to inactivate the enzyme. Cleaved products were visualized on a protein gel with controls of substrate without enzyme, and reaction at t = 0 minutes. MALDI-TOF (Bruker) was used to confirm the molecular weight of the cleaved product. Experiments were carried out at least on 2 batches.

### Statistics and reproducibility

Cell experiments in Fig. 4 and supplementary Fig. 15 were performed in triplicates twice with similar results. Target proteins were expressed at least three times and every batch was tested to confirm structure (CD, MS, SAXS) and similar activity. Data with only one replicate is representative of results seen throughout each batch's testing. No statistical method was used to predetermine sample size. The experiments were not randomized. The Investigators were not blinded to allocation during experiments and outcome assessment.

### Reporting summary

Further information on research design is available in the Nature Portfolio Reporting Summary linked to this article.

## Data availability

The datasets generated during and/or analyzed during the current study are available in the Figshare repository (https://doi.org/10.6084/m9.figshare.25351921)[96]. Sequencing data for SynIDPs can be found through GenBank accession numbers (PP454715, PP454716, and PP454717). NGS Data has been deposited to NCBI Sequence Read Archive and can be found with BioProject accession number PRJNA1090482. Source data are provided with this paper.

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

## Acknowledgements

This work was funded by the NIH through grant # R35 GM127042 to A.C. J.C.S. would like to thank the support of the NSF through a Graduate Research Fellowship. Y.S. would like to thank the support of the Zuckerman Foundation through woman in STEM award. We thank the Duke University School of Medicine for the use of the Sequencing and Genomic Technologies Shared Resource, which provided the next generation sequencing service. We also thank Prof. Terrence Oas for access to a circular dichroism spectrometer, and Dr. Peter Silinski for access to the MALDI-TOF mass spectrometer that was funded by the North Carolina Biotechnology Center Grant #2017-IDG-1018.

## Author contributions

N.C.T., J.C.S., and Y.S. performed experiments, analyzed data, and wrote the manuscript. G.K., S.D., P.S., and N.P. performed experiments. A.C. planned experiments, analyzed data, and wrote the manuscript.

## Competing interests

The authors declare no competing interests.
