## [Peer Review File · Nature Communications]

REVIEWER COMMENTS

Reviewer #1 (Remarks to the Author):

Review of Tang et al., submitted to Nature Communications

In this manuscript, the authors used two innovative approaches to improve recombinant expression of proteins in *E. coli*. 1) They used a pooled library of oligos to generate a large number of plasmids containing repetitive intrinsically disordered elements using rolling circle amplification. This is a challenging method, due to the repetitive elements most cloning methods would fail. This is an important accomplishment and should be shared to the community. 2) From this library, the authors screened for highly expressed soluble IDPs and showed their functionality in improving the folding of 3 recombinant proteins. Both of these accomplishments are novel enough to merit publication.

I have one major point and several points, that I would like the authors to address. I wish them success and look forward to seeing their results published soon.

Major Point:

- The authors claim that Z2LO10 and Tev are insoluble, but unlike TdT (Fig 3A) they do not show it. I would like to see how much of an improvement the SynIDP's accomplish. Did they make proteins 50% soluble or more or less?? Expression of Z2LO10 and Tev alone without SynIDP is necessary (my apologies if I missed it).
- Line 397: Authors claim that they show marked improvement of Tev in Fig S13A, but do not show it!! We only see the soluble Tev with the tag, no data without it.
- Negative controls without SynIDP is missing in Fig 3D and 3F.
- The authors have clearly demonstrated the utility of SynIDP's ability to improve folding of their cargo, but have not compared it to the common solubility tags such as MBP, NusA and SUMO. Comparing SynIDPs to these other commonly used tags would be extremely useful for the reader to understand whether they should try these tags in their next expression studies, or keep using the common ones.

Minor Points:

- Line 28: Change "Furhermore," to "Furthermore,"

- Line 29: Change “, but do not” to “may not always”, as some do. Many scientists have demonstrated improved folding due to MBP fusion, by showing active soluble cargo protein after removal of the MBP fusion tag.
- Line 30: Maybe change “need to design proteins” to “design solubility tags” as some are not proteins?
- Line 30: Recommend changing “There is hence a clear need” to “Hence, there is a clear need”
- Line 32: There is an abrupt transition from the first paragraph describing the available solubility tags to the second paragraph which states “To explore the hypothesis that polypeptide tags derived from intrinsically disordered proteins (IDPs) can promote the soluble expression of recombinant proteins...” I would recommend adding a sentence or two to introduce the idea of using IDPs as tags.
- Line 54: Repetition of Line 34 “hereafter”
- Line 58: Change “We hence” to “Hence, we”.
- Line 62: Change “and may hence serve as a useful tag” to “and therefore may serve as a useful tag.”
- Line 67: I do not understand what is “heuristic” about the authors approach. Please clarify.
- Line 73: “enormously large” is redundant. Recommend using just one adjective here.
- Lines 79, 86,102, and Fig1A legend: use different numbers for the 72-mer library. 1020, ~1000, and 2024. Please use one number and stay consistent for clarity.
- Line 89: Change “We designed the 72 nucleotide sequences” to “We designed the 72 nucleotide long sequences” to clarify that it is not 72 sequences, but the 1020 sequences that are 72 nt long.
- Lines 91, 114, 345: Change 5'- and 3' to prime symbols.
- Line 101: *in vitro* should be fully in italics.
- Line 104: Please clarify what makes the primer exonuclease resistant.
- Line 106: 5-methyl cytosine should be written 5-methylcytosine.
- Line 112: This is a full digestion of the DNA by SexAI, it just does not cleave methylated DNA. Partial digestion implies a reduced digestion time or units of enzyme.
- Line 122: Is MSKGP the result of SexAI cleavage site? Please clarify.
- Paragraph line 134-148: Maybe it was stated somewhere else, but could the authors state somewhere in this paragraph how many theoretical sequences they synthesized and out of that theoretical number what the final number they sequenced? Its not clear in this paragraph. Does 3888 unique alignments mean unique SynIDP's?
- Line 145: Another limitation is toxicity. Sequences that are toxic were also not selected.
- Line 158: I am confused about “filtered proteins”. I thought the colonies were transferred to nitrocellulose paper, no filtering involved??? Please clarify.

- Line 162: How do the authors know that dark spots are soluble? Antibodies against His-tag can also bind to inclusion bodies.
- Fig1A: Low resolution and text not clear. Show ladder size of the gel. Which part of the amplification was size selected? Please indicate whether it was the lower sized bands or the higher sized.
- Fig1C: Again, the text is not clear.
- Is there any example in nature of the existence of these SynIDP's in nature? Maybe a small section of a disordered region? That might be interesting and if there is none, the authors should state that too.
- Lines 195-196: Sentence wording is confusing. Suggest rewording to : "E. coli BL21(DE3) cells harboring a SynIDP encoding plasmid"
- Line 236: Change "All proteins also..." to "These proteins also..."
- Lines 250-251: Significant is used twice in this sentence.
- Line 257: Modify sentence "The transformed cells were expressed overnight..." The protein is expressed overnight, not the cells.
- Line 279-284: What is the significance of their findings? To those unfamiliar with this technique, what does the lack of LCST signify? Maybe a descriptive clarification sentence is necessary here. Line 339: SynIDP seems to behave different than the other two. Maybe in the discussion cumulate all the differences and explain with the mutations that add rigidity??
- Line 341: Why state the authors were unable to purify 'small amount' of TdT? Shouldn't they simply state that they were not able to purify TdT, leaving the 'small amount' out?
- Fig 3B: What is in the lane 3, no label (I'm assuming its SynIDP3-mTdT)
- Fig 3C: What is prTdT? Labels at the base of the figure are intersecting with the gel image and should be moved down.
- Figure 3G: All the x axis labels should be aligned for clarity. In the figure caption some of the m/z ratios have commas and some do not. Formatting should be consistent.
- Line 450: Add ref 22 to the ref 23.
- Line 451: Cite examples of toxic IDP's.
- Line 458: promote is spelled with an e.
- Line 464: SHuffle is with a capital H.

Reviewer #2 (Remarks to the Author):

Main results and significance

The manuscript by Tang et al, "Synthetic intrinsically disordered protein fusion tags that enhance protein solubility", presents a novel and ingenious method for synthesising a library of disordered protein coding sequences suitable for use as solubility and folding tags. The cost of synthesising these molecules, defined as SynIDP, is significantly lower than more conventional synthesis methods, or even compared to innovative methods already developed by Prof. Chilkoti's group.

The work presents the solution of a series of technical problems ranging from the design of the sequences, to the generation of the library, to the development of a highly selective screening. The whole process resulted in three tags capable of significantly improving the folding and solubility of three model proteins, TdT, TEV and Z2-LO10, known to form inclusion bodies when expressed in *E. coli* cells. These new solubility tags may not need to be removed from the target protein, allowing it to manifest its biological activity.

General considerations

Overall, the methodology is sound and the experimental results are in support of the conclusions. The text is generally understandable, although uneven in the degree of details and clarity (see below). In particular, the presentation of experimental design and the discussion could be improved. I find the quality of the figures and labels to be often poor.

1) Design of tag sequences - abundance of charged residues. In their tag library, the Authors reproduce two salient aspects of natural IDPs: the absence of secondary structure and the repetition of elements of low sequence complexity. On the other hand, they overlooked the low frequency of aromatic/hydrophobic residues and the high abundance of charged residues, which are critical elements of the composition of IDPs/IDRs. What is the rationale of this choice? I encourage comments on this issue, especially in light of the G₁₀E mutant, which shows greater solubility compared to the originally designed tags.

2) Design of tag sequences - tag length/size. The size of novel IDP-based solubility tags is not the result of a free directed evolution process, but reflects specific experimental constraints.

What considerations led the Authors to choose the 15-20 kDa size? Authors should clarify and discuss this aspect of their experimental design.

In this regard, it is worth recalling that Santner et al. (2012; ref. 23; doi: 10.1021/bi300653m) had already hypothesized that an effective solubility tag should have adequate dimensions compared to the dimensions of the target protein for entropic reasons, being able to "brush" the space around to avoid aggregogenic interactions in measure of its excluded volume.

In a subsequent publication, Tedeschi et al. (2017; doi: 10.1016/j.bbagen.2017.09.002) have hypothesized that a "charged" solubility tag, regardless of its size, could serve to nullify/reduce the net charge per residue (NCPR) of target proteins prone to aggregate at their pI. This reasoning suggests that size might be less important than composition, and is reminiscent of the Authors' reasoning on the efficacy of most hydrophilic tags to solubilize the most hydrophobic target proteins, such as mTdT. In line with this, one could perhaps hypothesize that even much smaller polypeptides (a few dozen bases?) are effective as solubility tags. The Authors could enrich the discussion with an explanation of their point of view on this issue.

3) Experimental results. The text is uneven in its level of detail and clarity. Three types of issues: i) in some cases, method/operational details seem useless or misleading; ii) some passages remain decidedly obscure due to the lack of general information on the method, such as that of CoFi; iii) some passages indulge in excessive comments that are not necessary in the "Results" and deserve to be moved to the "Discussion".

4) Discussion: As indicated above, the Authors are encouraged to attempt to delineate whether a general rule for the construction of an effective solubility tag emerges from their heuristic /empirical approach. Is it conceivable that the type of tag emerged from this work is well suited to the folding/solubilization of proteins with a pronounced hydrophobicity?

In this line, one could consider the possibility of rational design based on the Urry scale. The Authors could also consider the possibility of increasing the diversity of constructs by mutagenic RCA procedures (see for instance: Fujii R, Kitaoka M, Hayashi K. Error-prone rolling circle amplification greatly simplifies random mutagenesis. *Methods Mol Biol.* 2014;1179:23-9. doi: 10.1007/978-1-4939-1053-3_2)

Detailed comments/suggestions

Line 89: "We designed the 72 nucleotide sequences to contain exactly one non-palindromic methylation-sensitive SexAI recognition site to overlap with the Pro-Gly coding sequence that is split between the 5'- and 3'-ends of the oligonucleotide sequence (Figure S1)."

I would use the form "72-nucleotide sequences". The statement that the SexAI recognition site "is split between the 5' and 3' ends of the oligonucleotide sequences" remains completely unclear at this point in the description of the experimental design. The reference to Figure S1 does not help. I recommend deleting this sentence and just keeping the explanation given at line 114.

Line 102: "A pool of ~1000 linear 72 nucleotide (nt) long ssDNA templates were circularized via CirLigase II (Epicentre (Lucigen), USA)."

I would guess that the entire pool (i.e. 1020) of linear 72-nucleotide ssDNA molecules has been circularized.

Line 104: "Exonuclease resistant primer was annealed to the circular templates prior to isothermal polymerization at 30 °C using ϕ 29 DNA polymerase, a highly processive and strand displacing polymerase."

I imagine that the "exonuclease resistant primer" is actually a mixture of oligonucleotides capable of random priming on circularized ssDNA. If it is a commercial product, the Authors should indicate the source, otherwise they should specify its features and what causes its exonuclease-resistant properties.

Line 130: Following overnight growth, all colonies were combined by scraping with 1 mL of fresh media, and the plasmid DNA were purified by a plasmid isolation kit in a single tube, and prepared for next generation sequencing (NGS, see experimental). I would say that "colonies were collected by scraping..."

Line 153: "Plates were replica plated on additional LB-kanamycin plates..." I find not clear the description of the CoFi method. In the "Results", the Authors could shorten the entire description of the method by keeping just its objective (line 161: "The CoFi method positively screens for soluble proteins at the colony level"). The description of the CoFi procedure could be moved to the Materials and Methods and, to improve the accessibility of the paper, accompanied by a new supplementary figure illustrating the method schematically - this may be optional.

Line 175: "Next, we used PCR amplification directly from the E. coli colonies that showed soluble expression in CoFi to screen and eliminate from further consideration short genes of undesired lengths that are not removed by the CoFi method."

The CoFi method is not expected to discriminate clones expressing short proteins. Only if there is a correlation between short proteins and a propensity for insolubility should CoFi have contributed to the removal of shorter SynIDPs.

On the other hand, why are relatively short tags avoided? What is the length limit (20 repeats?) and what would determine it?

Line 177-182: These considerations on the need for further selective steps can be much more succinctly stated here and perhaps further explored in the Discussion.

Line 184: "At this stage, duplicates of different lengths were removed, and the remaining colonies were labeled so that their sequences could be tracked during protein expression. A total of 52 parent motifs (Table S2) were conserved, where some sequences included minor missense mutations, which slightly changes the overall amino acid composition of the parent motif."

There are several elements in this passage that deserve explanation. As I understand it, "parent motifs" group together conserved polypeptide sequences. Could the Authors quantify the degree of similarity between sequences belonging to the same motif in terms of sequence identity or similarity?

What is meant by "colony labelling"? How was it done and for what purpose?

Lines 193-196: "To facilitate a high throughput production process (HTPP), we used the Duetz Microflask system to cultivate cells with high growth rates and protein yields, at a throughput of about 48 SynIDP candidates per expression round.

The detail on sample size (48) raises questions about production in the Microflask system. In fact, it draws attention to the fact that the HTPP system can process (approximately?) 48 of the 52 samples derived from the previous screening steps. What happened to the 4 champions that are missing to complete the set of 52? How many rounds of screening were carried out? However, I would delete this information from the "Results" and move it to "Material and Methods" section. Further, I noticed that the acronym "HTPP" has only been used once, at the time of its introduction, and is therefore useless.

Lines 196-203: "Following growth of the E. coli BL21(DE3) cells harboring a plasmid for a SynIDP for 24 h, cells were harvested for purification. Soluble and insoluble fractions of the cell lysate were separated by centrifugation at 2800 g for 30 min at 4 °C, and screened for protein expression..... Visualization of the blot assay showed distinct dots (Figure 1E, Figure S5) for the soluble and insoluble fractions."

I would avoid here to specify the conditions for separating soluble and insoluble fractions, also because the cell lysis procedure is completely omitted. I suggest to keep this kind of information in the Materials and Methods section.

If the Authors have analysed both fractions, soluble and insoluble, it is obvious that two sets of dots appear. Please rephrase.

Lines 204-206: "Candidate SynIDPs for subsequent analysis were selected by the following criteria: (1) Soluble expression level in CoFi; (2) Soluble expression level in dot blot and PAGE gels; (3) Homogenous DNA plasmid population".

It is not clear to me if SDS-PAGE served to further screen the size of highly soluble SynIDPs. The gels in Supplementary Figure S4 do not have a label indicating the standard MW, and hence that of selected tags.

Lines 244-247: "SDS-PAGE of insoluble and soluble fractions of these proteins (Figure S6A) expressed in BL21(DE3) E. coli at 37 °C showed that TEV and Z2-LO10 are indeed insoluble. However, there was no detectable expression of mTdT at 37 °C, even as insoluble protein. As previous studies have reported expression of mTdT at lower temperatures⁵¹, we attempted its expression at 16 °C."

The text does not accurately reflect the SDS-PAGE shown in Supplementary Figure S6A. In fact, I would say that TEV and Z2-LO10 are largely produced in insoluble form, while sample overloading does not rule out their presence among the soluble proteins. In the case of mTdT induced at 37°C, a band of insoluble

protein is still clearly visible, being greatly increased when the induction temperature is reduced from 37°C to 16°C.

Lines 277-279: "Circular dichroism (CD) spectroscopy at 37°C (Figure 2B) showed that the SynIDPs lack defined secondary structure, as they have a negative peak at ~197 nm and a positive peak at ~215 nm that are characteristic of a random coil 29,34".

Can differences in the three CD spectra be attributed to different degrees of disorder? Have the Authors applied disorder predictors to the three proteins?

Lines 321-323: "Given that the Urry scale is derived from measurements on SynIDPs, we hypothesized that it would provide a predictive strength for assessing the hydropathy of the disordered proteins in this work."

I find this sentence not clear

Lines 363-367 "Interestingly, when using double the SynIDP-mTdT concentration and 1:500 initiator:dTTP ratio, we were able to stoichiometric incorporation"

It is not clear whether "double amount" means 26.4 µM. Further, to compare SynIDP-mTdT with the commercial enzyme, the amounts of the two enzyme should be also indicated and compared.

Lines 429-430. "The primary costs consisting of the initial purchase of oligonucleotides (~\$1k per 430 library) and DNA sequencing."

Not clear. The main verb is missing.

Lines 432-433: "Furthermore, this unique RCA based approach for the cloning purposes of synIDPs proved useful to evolve beneficial mutations that are otherwise extremely hard to incorporate."

In relation to the issue of introducing mutations into the nucleotide sequences of SynIDPs, the following work could be cited: Fujii R, Kitaoka M, Hayashi K. Error-prone rolling circle amplification greatly simplifies random mutagenesis. *Methods Mol Biol.* 2014;1179:23-9. doi: 10.1007/978-1-4939-1053-3_2.

Lines 443-444: "We also showed that scoring the hydropathy of the SynIDP sequence using Urry hydropathy scale is useful computational tool to predict the solubility of SynIDPs."

Could the opposite approach be taken? That is, could site-directed mutagenesis of previously selected sequences identify a sequence that optimizes the score?

Line 458: “promote” instead of “promot”

A few details

The headings of the "Results" paragraphs have a rather heterogeneous style (complete sentences with active verb; nouns with specification complements; nouns and prepositions indicating experimental purposes...). I recommend a more homogeneous style.

Throughout the text, but especially in “Materials and Methods”, the centrifugation conditions are given in a heterogeneous way, i.e. rpm or k RCF. Again I suggest using a more homogeneous way, for example, RCF, using the number of "g".

In some cases synthetic IDPs are referred to as synIDPs, instead of SynIDPs. Please standardize the wording if not linked to a specific meaning.

The origin of materials and equipment is recorded unsystematically.

Reviewer #3 (Remarks to the Author):

Tang and coworkers present a manuscript in which they describe the design and selection of intrinsically disordered fusion tags that enhance the solubility and expression of proteins while not interfering or even restoring protein function.

To preface – I am not an expert in protein expression and cannot comment on the originality and usefulness of the solubility tags developed here. However, the idea of the paper seems plausible. The paper is well written and understandable and approachable to researchers from adjacent fields. The Underlying hypotheses and research rationale are very clearly stated. I enjoyed reading the manuscript and would anticipate that it would be useful to the broad readership of Nat. Commun.

I was asked to comment specifically on the gene library generation and RCA amplification, which look plausible and experimental procedures described in enough detail to reproduce the experiments. The idea of creating pooled repetitive genes from an oligonucleotide library by RCA and partial digest is a clever solution for the generation of IDPs.

A few minor comments:

In their abstract they claim that “Novel DNA synthesis technology” was used. DNA microarrays were first developed in the late 1980s and have been commercially available since the early 90’s, but maybe that is not what they mean. The authors need to specify what exactly is novel and summarize their RCA/partial digest approach in a sentence or two.

Is it known if the methylated C is influencing RCA efficiency or cloning in any significant way?

Table S1 does not show oligonucleotides as it says in the caption but peptides. The actual DNA sequences from the oligonucleotide library should be added as the choice of codons might be important.

Font sizes in figures are sometimes too small (e.g. Fig 1 A, C) and the gel in 1A is too small to recognize anything. Is that just a space holder for 1B? Maybe indicate (with an arrow or such) that the bands were physically cut out of the gel and extracted.

Many gel images are too dark for many screens or when printed. Adjust brightness and contrast such that the gel background is a light shade of gray while avoiding overexposing the bands. (3 D, F; S4 bottom; S6, S15)

Table 1 could be compressed in size; the values of the Urry scale should be restricted to significant digits.

In the summary or in a figure, the best tags and their DNA sequences could be highlighted for readers seeking to adapt the SynIDPs for their own research.

Reviewer #4 (Remarks to the Author):

The authors created a large library of repetitive genes that encode synthetic intrinsically disordered proteins (synIDPs), and they screened the gene products to discover highly soluble ones that can be used as solubility-enhancing fusion tags in *E. coli*. The utility of the newly identified synIDPs was validated by showing that three aggregation-prone proteins (mTdT, Z2-LO10, and TEV protease) fused to the synIDPs were expressed in soluble forms without functional destruction. Although the approach for the generation and screening of the IDP sequence library is impressive, I’m not convinced that this study represents a major advance in the field of recombinant protein expression.

First, the most critical point is that no comparison data on target protein expression was provided with the other commonly used solubility-enhancing fusion tags, such as MBP, NusA, or SUMO. In addition, the authors justified the need for the design of synIDPs by mentioning that protein expression by using

conventional fusion tags can result in soluble but inactive forms of proteins (lines 28-31). This justification would be valid only when the synIDPs can promote the functional folding of a target protein that is otherwise inactive even with a fusion of conventional tags. By demonstrating an improved performance of the novel synIDP fusion tags compared to those of the conventional fusion tags, the usefulness (or powerfulness) of the synIDPs may be recognized indisputably.

Second, the originality of the study is overclaimed. It has already been documented that IDP tags can promote the soluble expression of recombinant proteins in at least four different papers [1-4]. However, the authors claim that they 'hypothesized' such an effect of IDP tags, without any rationale (lines 32-33 and 52-53). In addition, they cited the reference [1] as the only one previous paper that explored and used IDPs as a tag to promote soluble protein expression (lines 448-450). However, as pointed out, there are at least three more papers that demonstrated the use of IDP as a solubility-enhancing fusion tag, of which one paper [3] is already included in the reference list in the manuscript.

References

[1] *Biochemistry* 51, 7250–7262 (2012) (reference 23 in the manuscript)

[2] *Biomacromolecules* 15, 1194–1203 (2014)

[3] *Appl. Environ. Microbiol.* 88, e0009722 (2022) (reference 22 in the manuscript)

[4] *J. Microbiol.* 60, 960–967 (2022)

The following are more specific issues that must be addressed.

-Lines 56-58: The second reason why 'synthetic' IDPs, instead of native IDPs, are required is less convincing. Because recombinant proteins are mostly expressed in a heterologous host, potential interference of a native IDP with cellular function or metabolism would not be a problem. Conversely, a synthetic IDP might have a potential adverse effect on the cellular function or metabolism of the host.

-Line 58: The term 'SynIDPs' was already mentioned in line 34.

-Lines 255 and 334: Which one is correct, SynIDP-tev-H6 or SynIDP-H6-tev?

-Figure S7: The Western blot data on murine TdT (mTdT) expression is weird. In the lysate, the sizes of mTdT fused to the SynIDPs are all around 73 kDa, but in the soluble fraction, the sizes are shifted to around 59 kDa. It cannot be due to the cleavage of SynIDPs from the full-length proteins because, in that case, the 59 kDa bands must also appear in the lysate with the same band intensities observed in the soluble fraction. I cannot understand this result, but there is no explanation for this important issue.

-Figure S8: The band position of SynIDP-3 on the gel (above 20 kDa marker protein) was shifted downward (below 20 kDa and close to 15 kDa) after SEC. Please provide any explanation for this result.

-Figure 3A: The band positions of mTdT proteins are incorrect. The untagged mTdT should appear around 59 kDa.

-Throughout the manuscript, the authors claimed that the target proteins were functional without cleaving off the SynIDP tags. This might be beneficial in terms of production cost. However, during IMAC (Figure S11A), most of the recombinant proteins appeared in the flow-through fraction, which indicates inefficient binding of the proteins to the Ni-NTA resin. The tethered SynIDPs might interfere with the binding. In this situation, even though the SynIDPs could improve the soluble expression of the target protein, the final yield of purified protein would be very low, severely reducing the cost-effectiveness of protein production. In this regard, the cleavage of SynIDP tags seems necessary to improve the final yield of the target protein. In addition, although the TEV cleavage site was inserted in the constructs with SynIDPs, it has never been used (cleaved) in this study! Therefore, I suggest further experiments on the cleavage of SynIDP tag from SynIDP-tev-H6-mTdT prior to the protein purification by IMAC to see whether the tag-cleaved H6-mTdT can be highly purified without any binding interference from the SynIDP tags. For this purpose, I recommend the use of both commercially available TEV protease and SynIDP-TEV protease constructed in this study to compare the performances on the same substrate.

-Figure S12B: For SynIDP-1-Z2LO10, the two lanes for W2 and E seem to be mislabeled to each other.

-Lines 452-455: The authors claimed that they expressed and purified active mTdT using a fusion tag for the first time. Regardless of whether it is really the first study, I do not understand why this is important. As the authors have used in this study, recombinant TdTs from calf thymus are already commercially available. The activity of SynIDP-mTdT seems even lower than that of the commercially available enzyme (lines 363-366). Are there any beneficial characteristics of mTdT that are distinct from other TdT isozymes?

Reviewer comments are provided below in *italics*, and our responses are shown below each comment in blue. Manuscript text is shown in Arial font with specific sections added being highlighted.

We thank the reviewers for their time and effort providing us with such detailed comments, and our edits in response to these comments have improved the quality of our manuscript.

Reviewer #1

- The authors claim that Z2LO10 and TEV are insoluble, but unlike TdT (Fig 3A) they do not show it. I would like to see how much of an improvement the SynIDP's accomplish. Did they make proteins 50% soluble or more or less?? Expression of Z2LO10 and TEV alone without SynIDP is necessary (my apologies if I missed it).

Expression of Z2LO10 and TEV alone without SynIDP is shown in figure S8A, S15 and S16.

- Line 397: Authors claim that they show marked improvement of Tev in Fig S13A, but do not show it!! We only see the soluble Tev with the tag, no data without it.

Expression of TEV alone without SynIDP is shown in figure S8A. However, to make the manuscript more straightforward for the reader, we have added a new panel to Figure S16, which now shows the total and soluble expression of TEV alone, soluble expression of SynIDP-TEV and IMAC of SynIDP-TEV.

Figure S16: Purification of SynIDP-TEV. SDS-PAGE visualizes total and soluble expression of TEV alone (A) soluble expression of SynIDP-TEV (B). Representative SDS-PAGE from IMAC of SynIDP2-TEV (C) FT = flow through, W1 = wash with PBS buffer + 50 mM imidazole, E = elution with PBS buffer + 500 mM imidazole. The bands marked by arrows indicate the locations of target proteins.

- Negative controls without SynIDP is missing in Fig 3D and 3F.

Expression of Z2LO10 and TEV alone without SynIDP is shown in figure S8A, S15A and S16A (added).

- The authors have clearly demonstrated the utility of SynIDP's ability to improve folding of their cargo, but have not compared it to the common solubility tags such as MBP, NusA and SUMO. Comparing SynIDPs to these other commonly used tags would be extremely useful for the reader to understand whether they should try these tags in their next expression studies, or keep using the common ones.

We thank the reviewer for the thoughtful suggestion, which we carried out as suggested. In short, MBP-mTdT and SUMO-mTdT were expressed and the enzymatic activity of the purified proteins was compared to that of SynIDP1-mTdT.

The results indicate that although MBP and SUMO can produce soluble mTdT when fused on the N-terminus of the protein, their contributions to solubilizing their fusion partner in this case is less than that of SynIDP-1 (Figure 3D). Additionally, when testing the activity of SynIDP1-mTdT compared to that of SUMO-mTdT and MBP-TdT, the degree of polymerization of a poly T₅₀ initiator was higher with SynIDP-1-mTdT and the polydispersity lower compared to SUMO and BMP fusions of mTdT.

This data was incorporated into the manuscript, which now reads (p. 15):

To compare the performance of SynIDP with existing solubility tags, we expressed His₆-MBP-His₆-mTdT and His₆-SUMO-His₆-mTdT under the same conditions. Visualization of soluble and insoluble fractions of SynIDP-1-mTdT, SUMO-mTdT and MBP-mTdT (Figure 3D) on SDS-PAGE demonstrates significant enrichment of the target protein in the soluble fraction of SynIDP-1-mTdT compared to soluble fractions of SUMO-mTdT and MBP-mTdT. SUMO-mTdT and MBP-mTdT also exhibited a larger fraction of insoluble mTdT expression than SynIDP-1-mTdT, highlighting the improvement in soluble expression of mTdT when fused to SynIDP-1 compared to SUMO and MBP. In addition to the constructs described above, a cleaved mTdT was generated via incubation of the soluble fractions of SynIDP-1-mTdT with SynIDP-1-TEV (further described in the following sections) at 4 °C overnight. The constructs were purified using IMAC (Figure S14A, B) and SEC.

Enzymatic activity of the mTdT variants was measured by TcEP of ss-DNA (Figure 3E, Figure S14D). To that end, all mTdT variants were buffer exchanged into the optimal reaction buffer (50 mM Potassium Phosphate, 100 mM NaCl, 1 mM 2-mercaptoethanol, 0.1% Tween 20, 50% glycerol, pH 6.4) to a final concentration of 0.5 μM (Figure S14C) or 1 μM (labeled as 1X or 2X). Recombinant, soluble mTdT demonstrated very little activity (Figure S14D) as mentioned above. In contrast, all of the mTdT variants expressed as fusions to solubility tags, as well as the cleaved mTdT from SynIDP-1-mTdT demonstrated a marked improvement in nucleotide addition compared to recombinant mTdT. This increase in enzymatic activity is also interesting to note that removal of the SynIDP-1 tag from mTdT results in loss of protein activity. Further analysis of the TcEP products was performed by image processing of the original gel (Figure S14D, Figure 3E) to obtain the fluorescence intensity distribution for each of the mTdT variants (Figure 3F) as function of distance from the gel bottom. For the TcEP reaction, we were interested in obtaining a high degree of polymerization, which is indicated by a higher distance from the bottom of the gel, and low dispersity, which can be evaluated by the width of the fluorescence signal. Comparing the fluorescence intensity curves of the mTdT variants shows that SynIDP-mTdT outperforms the other mTdT variants; SynIDP1-mTdT shows the highest

degree of polymerization and the lowest dispersity (Figure 3E, 3F), indicating the superior folding of mTdT within the cohort of solubility tags tested.

Figure 3. Fusion to SynIDPs rescues soluble and functional expression of mTdT. **A.** Western blot of insoluble and soluble fractions of mTdT and SynIDP-1/2/3-mTdT using anti-TdT antibody. **B.** Purified SynIDP-1/2/3-mTdT after IMAC and SEC visualized on SDS-PAGE. **C.** mTdT activity assay showing elongation of Cy5-poly-T₅₀ initiator on TBE-Urea PAGE gels. Low Range ssRNA ladder was used to quantify nucleotide addition. From left to right; ladder, initiator (negative control), Promega-TdT (positive control), SynIDP-1-mTdT 1X and 2X, SynIDP-2-mTdT 1X and 2X, SynIDP-3-mTdT 1X and 2X. **D.** Insoluble (Ins) and soluble (S) fractions of mTdT, SynIDP-1-mTdT, SUMO-mTdT and MBP-mTdT visualized on SDS-PAGE. Large increase in soluble expression of mTdT are observed when fused to SynIDP-1 compared to fusion with SUMO or MBP. Arrows indicate the desired protein product. **E.** TdT activity assay showing elongation of Cy5-poly-T₅₀ initiator on TBE-Urea PAGE gels. Low Range ssRNA ladder was used to quantify nucleotide addition. From left to right; cleaved mTdT, SynIDP-1-mTdT, SUMO-mTdT, MBP-mTdT, Ladder. Image was processed from Figure S14D (see experimental). **F.** ImageJ analysis of fluorescence intensity as measurement for dispersity and degree of polymerization of mTdT variants' reaction products. The peak at a distance of ~100 pixels stems from the dye front and is not a real product. SynIDP-1-mTdT outperforms the other variants by displaying lower dispersity and a higher degree of polymerization. The corresponding nt size to the distance (shown in the line above) was obtained by similar analysis of the ladder.

And in Supplementary Information (p.55):

Figure S14: Purification of cleaved mTdT, SUMO-mTdT and MBP-TdT. (A) MBP-mTdT and SUMO-mTdT purification by IMAC visualized by SDS-PAGE. We note that MBP-His₆-mTdT and SUMO-His₆-mTdT (no His₆ tag at the terminus) were impossible to purify by IMAC **(B)** Soluble fractions from SynIDP-1-mTdT and SynIDP-1-TEV were combined in a 1:1 v:v ratio and incubated at 4 °C overnight. The resulting mixture was purified by IMAC and visualized on SDS-PAGE. The desired products are indicated by arrows. Ins = insoluble fraction, S = soluble fraction, FT = flow through, W1 = wash with Lys buffer + 25 mM imidazole, W2 = wash with Lys buffer + 50 mM imidazole, E1 = elution with Lys buffer + 100 mM imidazole, E2 = elution with Lys buffer + 500 mM imidazole. **(C)** Purified mTdT variants after IMAC and SEC. **(D)** TdT activity assay showing elongation of Cy5-poly-T₅₀ initiator on TBE-Urea PAGE gels. Low Range ssRNA ladder was used to quantify nucleotide addition. From left to right; SynIDP-1-mTdT 2X and 1X, recombinant m-TdT 2X and 1X, SUMO-mTdT 2X and 1X, MBP-mTdT 2X and 1X, cleaved mTdT 2X and 1X, NEB-TdT (positive control), initiator (negative control), ladder. The assay indicates that recombinant mTdT has very little activity. In contrast, the cleaved mTdT as well as all the mTdT variants expressed using a solubility tag retain enzymatic activity, with SynIDP-1-mTdT demonstrating the highest processivity and lowest dispersity (Figure 3F).

For the next 17 language-related comments we have made the changes exactly as suggested.

- Line 28: Change “Furhermore,” to “Furthermore, ”
- Line 29: Change “, but do not” to “may not always”, as some do. Many scientists have demonstrated improved folding due to MBP fusion, by showing active soluble cargo protein after removal of the MBP fusion tag.
- Line 30: Recommend changing “There is hence a clear need” to “Hence, there is a clear need”
- Line 30: Maybe change “need to design proteins” to “design solubility tags” as some are not proteins?
- Line 54: Repetition of Line 34 “hereafter”
- Line 58: Change “We hence” to “Hence, we”.
- Line 62: Change “and may hence serve as a useful tag” to “and therefore may serve as a useful tag.”
- Lines 79, 86,102, and Fig1A legend: use different numbers for the 72-mer library. 1020, ~1000, and 2024. Please use one number and stay consistent for clarity.
- Line 89: Change “We designed the 72 nucleotide sequences” to “We designed the 72 nucleotide long sequences” to clarify that it is not 72 sequences, but the 1020 sequences that are 72 nt long.
- Line 101: *in vitro* should be fully in italics.
- Line 106: 5-methyl cytosine should be written 5-methylcytosine.
- Lines 195-196: Sentence wording is confusing. Suggest rewording to : “E. coli BL21(DE3) cells harboring a SynIDP encoding plasmid
- Line 236: Change “All proteins also...” to “These proteins also...”
- Lines 250-251: Significant is used twice in this sentence.
- Line 341: Why state the authors were unable to purify ‘small amount’ of TdT? Shouldn’t they simply state that they were not able to purify TdT, leaving the ‘small amount’ out?
- Line 458: promote is spelled with an e.
- Line 464: SHuffle is with a capital H.
- Line 73: “enormously large” is redundant. Recommend using just one adjective here. Changed to “Enormous”.

- Line 32: There is an abrupt transition from the first paragraph describing the available solubility tags to the second paragraph which states “To explore the hypothesis that polypeptide tags derived from intrinsically disordered proteins (IDPs) can promote the soluble expression of recombinant proteins...” I would recommend adding a sentence or two to introduce the idea of using IDPs as tags.

We have changed the text according to the suggestion. The manuscript now reads as follows (p. 2):

It has been previously shown that disordered regions at the N terminus of folded native proteins promote solubility and prevent aggregation of proteins^{13, 14}. Due to lack of secondary structure, these disordered regions have high conformational freedom, and hence are believed to act as entropic bristles which prevent aggregation of the protein. Based on this knowledge, we hypothesized that polypeptide tags derived from intrinsically disordered proteins (IDPs) can promote the soluble expression of recombinant proteins.

- Line 67: I do not understand what is “heuristic” about the authors approach. Please clarify.

We apologize for the lack of clarity. The “heuristic” approach refers to loosely defined rules identified by the referenced authors (Garcia-Quiroz et al.) where P-G-X_n containing motifs encode disorder. These rules were shown to be robust by trial-and-error experiments of many different sequences and are therefore described as a heuristic. *The specific heuristic we used in this paper is choosing the sequence to be P-G-X₄ (to ensure no structure) and then narrowing down the 20⁴ DNA library by the requirement that X=G at least once and X≠C or P. In addition, G was excluded from the X₄ position.*

- *Line 104: Please clarify what makes the primer exonuclease resistant.*

We apologize for the lack of clarity. We used commercially available Exonuclease-Resistant Random Primer which is a mixture of single-stranded random oligonucleotides used for highly efficient random priming of various DNA synthesis reactions. This primer has two 3'-terminal phosphorothioate (PTO) modifications that are resistant to the 3'→5' exonuclease activity of proofreading DNA polymerases. It also has 5'- and 3'-hydroxyl ends. The manuscript now reads (p. 4):

An exonuclease resistant primer with 3'-terminal phosphorothioate (PTO) modifications (Thermo Fisher Scientific) was annealed to the circular templates.

- *Line 112: This is a full digestion of the DNA by SexAI, it just does not cleave methylated DNA. Partial digestion implies a reduced digestion time or units of enzyme.*

We have deleted the word “partial” throughout the manuscript.

- *Line 122: Is MSKGP the result of SexAI cleavage site? Please clarify.*

The MSKGP is a leading sequence present on the plasmid that SynIDP is cloned into.

- *Paragraph line 134-148: Maybe it was stated somewhere else, but could the authors state somewhere in this paragraph how many theoretical sequences they synthesized and out of that theoretical number what the final number they sequenced? Its not clear in this paragraph. Does 3888 unique alignments mean unique SynIDP's?*

Out of 2 million sequences returned by NGS, there were 3888 unique alignments based on reference oligonucleotides paired with compact idiosyncratic gapped alignment reports strings.

- *Line 145: Another limitation is toxicity. Sequences that are toxic were also not selected.*

Toxicity was added as a limitation and the manuscript now reads as follows (p. 5):

Sequence bias could arise from a variety of factors including, but not limited to, polymerase nucleotide insertion preference, preferential plasmid replication *in vivo* and toxicity.

- *Line 158: I am confused about “filtered proteins”. I thought the colonies were transferred to nitrocellulose paper, no filtering involved??? Please clarify.*

We apologize for the confusion. The colonies were transferred to nitrocellulose through a Whatman filter paper. The details are stated in the methods section.

To make this procedure clear to the reader, we have revised the description of the CoFi procedure and have added a new supplementary figure illustrating the method schematically. The manuscript now reads as follows (p. 6):

To rapidly identify soluble protein expression in *E. coli*, we implemented the Colony Filtration (CoFi) blot method (see experimental, Figure S3)³⁹. Briefly, a mix of plasmid DNA was electroporated into Acella™ electrocompetent BL21(DE3) *E. coli* and plated on LB-kanamycin plates. Plates were replica plated⁴⁰ on additional LB-kanamycin plates prior to transfer of the original plate's colonies on to a nitrocellulose membrane over a filter sandwich consisting of a Whatman paper drenched in lysis buffer. The resulting soluble blots were visualized by Western Blot with an anti-His₆ antibody (Figure 1D, Figure S4).

Figure S3: Illustration of the steps of the CoFi method. For a full description please see the methods section.

- Line 162: How do the authors know that dark spots are soluble? Antibodies against His-tag can also bind to inclusion bodies.

Inclusion bodies are retained on the Whatman paper in the transfer step, which is essentially a filtering step.

- Fig1A: Low resolution and text not clear. Show ladder size of the gel. Which part of the amplification was size selected? Please indicate whether it was the lower sized bands or the higher sized.

Text size in Fig1A was increased. The gel shown on the bottom of Fig1A refers to the same gel in Fig 1B and hence was replaced to make it clearer. The ladder size is shown and an arrow indicating the size of interest was added.

- Fig1C: Again, the text is not clear.

An enlarged image was added to supplementary information as Figure S2. The caption of Figure 1 was revised and now reads as follows:

Figure 1: The steps for identification of highly soluble SynIDPs. **A.** Generation of SynIDP gene library. A pool of 1020 ssDNA 72-nt long oligonucleotides were circularized by CircLigase and amplified using RCA. Substitution of dCTP with 10% 5-methyl-dCTP randomly incorporates 5mC into the RCA product, resulting in various DNA size products upon digestion by SexAI (Figure S1). The products can be separated by electrophoresis. Fragments of the desired size (360-576 bp) are then gel purified out and ligated into a plasmid. **B.** Restriction digest of RCA products with 0% 5mCTP (left) vs. 10% 5mCTP (right) using the restriction enzyme SexAI shows that partial digests occur in factors of 72 bp when 5mC is incorporated. Desired size (360-576 bp, marked with arrow) DNA are then gel purified out. **C.** Illumina Miseq analysis of cloned library of plasmid DNA verifies the presence of 865 unique motifs (See Figure S2 for enlarged version). **D.** CoFi protein expression (Figure S3) allows identification of *E. coli* colonies that express soluble protein by an anti-His₆ western blot (left panel). Colonies were identified manually on a greyscale image (middle panel) and a black and white pixel-thresholded image overlaying the green channel (right panel), was printed and aligned underneath the plate of colonies as a visual aid for colony picking. Images represent a subset of the total set of agar plates that were analyzed (Figure S4). **E.** Dot blot of insoluble and soluble lysate fractions to determine target protein solubility. The dot blot membrane was subjected to His₆-tag antibody detection. The distinct dots marked by arrows represent an example target protein, [GTHGTP]₂₄ that was determined to be soluble. Solubility was determined by the intensity of the soluble

- Line 257: Modify sentence “The transformed cells were expressed overnight...” The protein is expressed overnight, not the cells.

We have changed the wording and the manuscript now reads as follows (p. 10):

The transformed cells were **grown** overnight using overnight express TB.

- Line 279-284: What is the significance of their findings? To those unfamiliar with this technique, what does the lack of LCST signify? Maybe a descriptive clarification sentence is necessary here.

LCST behavior involves coacervation —phase separation— upon increasing the temperature above the cloud point temperature of the SynIDP fusion protein or upon increasing the salt concentration. Phase separation of SynIDPs can be useful to purify proteins as we have shown for ELP fusion protein (Meyer and Chilkoti, Nature Biotechnology, 1999) and was the basis for startup company founded by Chilkoti (isolerebio.com) but incorporating LCST phase behavior was not goal of this study and was not encoded in the design. We hence experimentally verified that this was indeed the case and have added descriptive clarification sentence to the manuscript, which now reads (p. 10):

...indicating a lack of LCST phase separation under these conditions, **ensuring no coacervation is expected due to elevated temperature or salt concentration.**

-Line 339: SynIDP seems to behave different than the other two. Maybe in the discussion cumulate all the differences and explain with the mutations that add rigidity??

Added to discussion. The manuscript now reads (p. 19):

The lack of secondary structure and the high conformational flexibility of the SynIDPs was validated using CD and SAXS, with SynIDP-3 having more rigid structure as compared to SynIDP-1/2 due to the presence of the G to E mutation every 6 repeats, effectively incorporating charge and repulsion along the chain.

- Fig 3B: What is in the lane 3, no label (I'm assuming its SynIDP3-mTdT).

Yes, it is SynIDP-3-mTdT. Label was added.

- Fig 3C: What is prTdT? Labels at the base of the figure are intersecting with the gel image and should be moved down.

prTdT refers to commercial Promega TdT-as stated in the caption. Labels at the base were moved further down.

- Figure 3G: All the x axis labels should be aligned for clarity. In the figure caption some of the m/z ratios have commas and some do not. Formatting should be consistent.

X axis labels were aligned. We formatted the m/z to be consistent.

- Line 450: Add ref 22 to the ref 23.

We have added the reference and revised the sentence, which now reads (p. 20):

We note that this is a **relatively** new and interesting application of IDPs, as there is **little previously published work** that explored and used IDPs as tags to promote soluble protein expression^{24,25,89}

- Line 451: Cite examples of toxic IDP's.

Ref 84 (doi.org/10.1016/j.bbrc.2006.01.151) describing examples of toxic IDPs was added to the manuscript (p. 20).

Reviewer #2

1) Design of tag sequences - abundance of charged residues. In their tag library, the Authors reproduce two salient aspects of natural IDPs: the absence of secondary structure and the repetition of elements of low sequence complexity. On the other hand, they overlooked the low frequency of aromatic/hydrophobic residues and the high abundance of charged residues, which are critical elements of the composition of IDPs/IDRs. What is the rationale of this choice? I encourage comments on this issue, especially in light of the G Δ E mutant, which shows greater solubility compared to the originally designed tags.

The only constraints for design of the SynIDP library were:

1. Sequence of P-G-X1-X2-X3-X4
2. X= Glycine at least once
3. X \neq Cysteine or proline
4. X₄ \neq Glycine
5. The SexAI restriction site, ACCWGGT, covers the last nt of X₄ followed by the codons of proline and glycine. The requirement of the SexAI restriction site imposes that X₄ must end with adenine, excluding the following amino acids from X₄: N, D, H, M, F, W and Y.

High abundance of charged residues and low frequency of aromatic/hydrophobic residues are theoretically covered in this SynIDP DNA library. The most soluble SynIDPs that have been experimentally identified in this work, indeed do not have any aromatic residue (as can be expected), however, they also don't have a high abundance of charge. We assume that SynIDPs that have high net charge might co-precipitate with other bacterial substances (such as DNA) and hence are not performing as highly soluble although having high hydrophathy. Similar results in regard to positive charge were seen by Chan *et al.* Discussion was added to the manuscript, which now reads (p. 19-20):

SynIDP-1/3 are good examples of the superiority of libraries and experimental screening for solubility over a rational design approach, **because** SynIDP-1/3 could not have been rationally designed for this purpose, as according to Wilkinson and Harrison solubility calculator^{69, 70} they would be expected to be insoluble. As an alternative calculator for predicting SynIDPs solubility, we suggest using Urry hydrophathy scale⁷¹. The most soluble synthetic IDPs that have been experimentally identified in this work, do **not** contain aromatic residues. This outcome can be expected, given that aromatic residues are well known **to promote** aggregation due to π stacking and hydrophobic interactions⁸⁸. **Counterintuitively**, the most soluble synthetic IDPs also lack positive charge. We speculate that SynIDPs that have high positive net charge **may** complex with other **positively charged macromolecules** such as DNA and precipitate out **even though they** have a high hydrophathy. Similarly, Chan *et al.* demonstrated that soluble expression of proteins correlates with a lack of positively-charged surface⁸⁹.

2) Design of tag sequences - tag length/size. The size of novel IDP-based solubility tags is not the result of a free directed evolution process, but reflects specific experimental constraints. What considerations led the Authors to choose the 15-20 kDa size? Authors should clarify and

discuss this aspect of their experimental design.

In this regard, it is worth recalling that Santner et al. (2012; ref. 23; doi: 10.1021/bi300653m) had already hypothesized that an effective solubility tag should have adequate dimensions compared to the dimensions of the target protein for entropic reasons, being able to "brush" the space around to avoid aggregogenic interactions in measure of its excluded volume.

In a subsequent publication, Tedeschi et al. (2017; doi: 10.1016/j.bbagen.2017.09.002) have hypothesized that a "charged" solubility tag, regardless of its size, could serve to nullify/reduce the net charge per residue (NCPR) of target proteins prone to aggregate at their pI. This reasoning suggests that size might be less important than composition, and is reminiscent of the Authors' reasoning on the efficacy of most hydrophilic tags to solubilize the most hydrophobic target proteins, such as mTdT. In line with this, one could perhaps hypothesize that even much smaller polypeptides (a few dozen bases?) are effective as solubility tags. The Authors could enrich the discussion with an explanation of their point of view on this issue.

We did not choose our IDPs to be in the 15-20 kDa range. For example, SynIDP3 is 13,805 Da. We were targeting our IDPs to be above 10 kDa, as in our experience expressing smaller SynIDPs leads to challenges in expression and purification. We avoided high molecular weights (>20 kDa) to avoid an unduly high metabolic burden on *E. coli* upon expression of the SynIDP-fusion protein. As our oligos are 72 nt (encoding to 4 repeats) the range of 20-28 repeats was the best to satisfy this criterion. It is also notable that one of the best performing IDPs from Santner et al. (ref 23) is also 15 kDa. We have also tried using an 8-mer (data not shown) with the same repeat sequence as SynIDP3 and it did not perform well as a solubility tag, suggesting that a minimum length is required.

3) *Experimental results. The text is uneven in its level of detail and clarity. Three types of issues: i) in some cases, method/operational details seem useless or misleading; ii) some passages remain decidedly obscure due to the lack of general information on the method, such as that of CoFi; iii) some passages indulge in excessive comments that are not necessary in the "Results" and deserve to be moved to the "Discussion".*

Revised to address these issues.

4) *Discussion: As indicated above, the Authors are encouraged to attempt to delineate whether a general rule for the construction of an effective solubility tag emerges from their heuristic /empirical approach. Is it conceivable that the type of tag emerged from this work is well suited to the folding/solubilization of proteins with a pronounced hydrophobicity?*

In this line, one could consider the possibility of rational design based on the Urry scale. The We have added some discussion about the possibility of rationally designing SynIDP as solubility tags. The manuscript now reads (p. 12):

Previous attempts to rationally design SynIDPs for the purpose of solubility tags used the Wilkinson and Harrison solubility calculator^{69,70} to predict the chance of soluble expression of a given protein in *E.coli*. For SynIDP-2, this calculator predicts 100% chance of solubility when overexpressed in *E.coli*. However, this calculator fails when calculating the solubility of SynIDP-1 and SynIDP-3, by predicting that there is 0% chance that these SynIDPs would be soluble.

(p19-20):

SynIDP-1/3 are good examples of the superiority of libraries and experimental screening for solubility over a rational design approach, because SynIDP-1/3 could not have been rationally designed for this purpose, as according to Wilkinson and Harrison solubility calculator^{69,70} they

would be expected to be insoluble. As an alternative calculator for predicting SynIDPs solubility, we suggest using Urry hydrophathy scale⁷¹. The most soluble synthetic IDPs that have been experimentally identified in this work, do not contain aromatic residues. This outcome can be expected, given that aromatic residues are well known to promote aggregation due to π stacking and hydrophobic interactions⁸⁸. Counterintuitively, the most soluble synthetic IDPs also lack positive charge. We speculate that SynIDPs that have high positive net charge may complex with other positively charged macromolecules such as DNA and precipitate out even though they have a high hydrophathy. Similarly, Chan *et al.* demonstrated that soluble expression of proteins correlates with a lack of positively-charged surface⁸⁹.

Authors could also consider the possibility of increasing the diversity of constructs by mutagenic RCA procedures (see for instance: Fujii R, Kitaoka M, Hayashi K. Error-prone rolling circle amplification greatly simplifies random mutagenesis. Methods Mol Biol. 2014;1179:23-9. doi: 10.1007/978-1-4939-1053-3_2)

Although increasing the diversity of constructs by mutagenic RCA procedures is an interesting idea, it is outside the scope of this paper.

Line 89: "We designed the 72 nucleotide sequences to contain exactly one non-palindromic methylation-sensitive SexAI recognition site to overlap with the Pro-Gly coding sequence that is split between the 5'- and 3'-ends of the oligonucleotide sequence (Figure S1)."

I would use the form "72-nucleotide sequences". The statement that the SexAI recognition site "is split between the 5' and 3' ends of the oligonucleotide sequences" remains completely unclear at this point in the description of the experimental design. The reference to Figure S1 does not help. I recommend deleting this sentence and just keeping the explanation given at line 114.

The manuscript now reads (p. 3):

We designed the 72-**nt long** sequences to contain exactly one non-palindromic methylation-sensitive SexAI recognition site to overlap with the Pro-Gly coding sequence (Figure S1).

Line 102: "A pool of ~1000 linear 72 nucleotide (nt) long ssDNA templates were circularized via CircLigase II (Epicentre (Lucigen), USA)."

I would guess that the entire pool (i.e. 1020) of linear 72-nucleotide ssDNA molecules has been circularized.

The manuscript now reads (p. 4):

The entire pool of 1020 linear 72-nt long ssDNA molecules was circularized via CircLigase II (Epicentre (Lucigen), USA).

Line 104: "Exonuclease resistant primer was annealed to the circular templates prior to isothermal polymerization at 30 °C using ϕ 29 DNA polymerase, a highly processive and strand displacing polymerase."

I imagine that the "exonuclease resistant primer" is actually a mixture of oligonucleotides capable of random priming on circularized ssDNA. If it is a commercial product, the Authors should indicate the source, otherwise they should specify its features and what causes its exonuclease-resistant properties.

The manuscript now reads (p. 4):

An exonuclease resistant primer with 3'-terminal phosphorothioate (PTO) modifications (Thermo Fisher Scientific) was annealed to the circular templates prior to isothermal polymerization at 30 °C using ϕ 29 DNA polymerase, a highly processive and strand displacing polymerase.

Line 130: Following overnight growth, all colonies were combined by scraping with 1 mL of fresh media, and the plasmid DNA were purified by a plasmid isolation kit in a single tube, and prepared for next generation sequencing (NGS, see experimental). I would say that “colonies were collected by scraping...”

The manuscript now reads (p. 5):

Following overnight growth, colonies were **collected** by scraping...

Line 153: “Plates were replica plated on additional LB-kanamycin plates...” I find not clear the description of the CoFi method. In the “Results”, the Authors could shorten the entire description of the method by keeping just its objective (line 161: “The CoFi method positively screens for soluble proteins at the colony level”). The description of the CoFi procedure could be moved to the Materials and Methods and, to improve the accessibility of the paper, accompanied by a new supplementary figure illustrating the method schematically - this may be optional.

The description of the CoFi method was shortened and we have added a new supplementary figure illustrating the method schematically. The manuscript now reads as follows (p. 6):

To rapidly identify soluble protein expression in *E. coli*, we implemented the Colony Filtration (CoFi) blot method (**see experimental, Figure S3**)³⁹. **Briefly**, a mix of plasmid DNA was electroporated into Acella™ electrocompetent BL21(DE3) *E. coli* and plated on LB-kanamycin plates. Plates were replica plated⁴⁰ on additional LB-kanamycin plates prior to transfer of the original plate's colonies on to **a nitrocellulose membrane over a filter sandwich consisting of a Whatman paper drenched in lysis buffer**. The resulting soluble blots were visualized by Western Blot with an anti-His₆ antibody (Figure 1D, Figure S4). The CoFi method positively screens for soluble proteins at the colony level. Dark spots on a His₆-tag Western blot indicate colonies expressing soluble full-length proteins (left panel, Figure 1D). These dark spots were manually identified for colony-selection from a greyscale image that was printed on paper (middle panel, Figure 1D). To facilitate the visual identification of previously selected colonies on the replica plates, the greyscale image was **processed** (right panel, Figure 1D). The resulting color image was properly scaled, printed, and fiducially aligned underneath the semi-transparent agar replica plates, facilitating subsequent colony picking. In this way, candidate SynIDPs for subsequent analysis were selected by their soluble expression level in CoFi.

Figure S3: Illustration of the steps of the CoFi method. For a full description please see the methods section.

Line 175: “Next, we used PCR amplification directly from the *E. coli* colonies that showed soluble expression in CoFi to screen and eliminate from further consideration short genes of undesired lengths that are not removed by the CoFi method.”

The CoFi method is not expected to discriminate clones expressing short proteins. Only if there is a correlation between short proteins and a propensity for insolubility should CoFi have contributed to the removal of shorter SynIDPs.

On the other hand, why are relatively short tags avoided? What is the length limit (20 repeats?) and what would determine it?

The reviewer is correct about CoFi not removing short SynIDPs, which is why PCR is performed to verify 20-28 repeats. The reasoning is stated above in response to the reviewer’s comment #2.

Line 177-182: These considerations on the need for further selective steps can be much more succinctly stated here and perhaps further explored in the Discussion.

We believe that leaving these lines as they will make the text more understandable to the reader.

Line 184: “At this stage, duplicates of different lengths were removed, and the remaining colonies were labeled so that their sequences could be tracked during protein expression. A total of 52 parent motifs (Table S2) were conserved, where some sequences included minor missense mutations, which slightly changes the overall amino acid composition of the parent motif.” There are several elements in this passage that deserve explanation. As I understand it, “parent motifs” group together conserved polypeptide sequences. Could the Authors quantify the degree of similarity between sequences belonging to the same motif in terms of sequence identity or similarity?

Summary of Sanger sequencing results, including motif, occurrences in NGS, occurrences in Sanger sequencing and number of mutations are shown in Table S2 in SI.

What is meant by "colony labelling"? How was it done and for what purpose?

Colony labeling refers to marking the colony with letter + number on the plate, as presented in Figure 1D middle panel. We refer the reader to the figure, and the manuscript now reads (p. 6): At this stage, duplicates of different lengths were removed, and the remaining colonies were labeled (Figure 1D middle panel) so that their sequences could be tracked during protein expression.

Lines 193-196: "To facilitate a high throughput production process (HTPP), we used the Duetz Microflask system to cultivate cells with high growth rates and protein yields, at a throughput of about 48 SynIDP candidates per expression round.

The detail on sample size (48) raises questions about production in the Microflask system. In fact, it draws attention to the fact that the HTPP system can process (approximately?) 48 of the 52 samples derived from the previous screening steps. What happened to the 4 champions that are missing to complete the set of 52? How many rounds of screening were carried out? However, I would delete this information from the "Results" and move it to "Material and Methods" section. Further, I noticed that the acronym "HTTP" has only been used once, at the time of its introduction, and is therefore useless.

The details on sample size in throughput as well as HTPP acronym were deleted. Several rounds of expression were carried out to accommodate for the excess of 48 candidates as well as some duplicate backup clones. Clones were then re-sequenced and consolidated prior to screening. The manuscript now reads (p. 7):

Several rounds of expression were carried out to accommodate multiple backup clones. Following growth of the *E. coli* BL21(DE3) cells harboring a SynIDP encoding plasmid for 24 h, clones were then re-sequenced to further confirm the absence of deleterious frameshifts or mixed populations, and the duplicates were then removed leaving single representative clones. Cells from the resulting clones were harvested for purification.

Lines 196-203: "Following growth of the E. coli BL21(DE3) cells harboring a plasmid for a SynIDP for 24 h, cells were harvested for purification. Soluble and insoluble fractions of the cell lysate were separated by centrifugation at 2800 g for 30 min at 4 °C, and screened for protein expression..... Visualization of the blot assay showed distinct dots (Figure 1E, Figure S5) for the soluble and insoluble fractions."

I would avoid here to specify the conditions for separating soluble and insoluble fractions, also because the cell lysis procedure is completely omitted. I suggest to keep this kind of information in the Materials and Methods section.

If the Authors have analysed both fractions, soluble and insoluble, it is obvious that two sets of dots appear. Please rephrase.

The specification of the conditions for separating soluble and insoluble fractions was deleted. Both figures show the full set of distinct dots. One labeled with names and one unlabeled.

Lines 204-206: "Candidate SynIDPs for subsequent analysis were selected by the following criteria: (1) Soluble expression level in CoFi; (2) Soluble expression level in dot blot and PAGE gels; (3) Homogenous DNA plasmid population".

It is not clear to me if SDS-PAGE served to further screen the size of highly soluble SynIDPs.

The gels in Supplementary Figure S4 do not have a label indicating the standard MW, and hence that of selected tags.

SDS-PAGE was not used to screen the size of SynIDPs but rather to compare soluble expression (left panel) to insoluble expression (right panel).

Lines 244-247: “SDS-PAGE of insoluble and soluble fractions of these proteins (Figure S6A) expressed in BL21(DE3) *E. coli* at 37 °C showed that TEV and Z2-LO10 are indeed insoluble. However, there was no detectable expression of mTdT at 37 °C, even as insoluble protein. As previous studies have reported expression of mTdT at lower temperatures⁵¹, we attempted its expression at 16 °C.”

The text does not accurately reflect the SDS-PAGE shown in Supplementary Figure S6A. In fact, I would say that TEV and Z2-LO10 are largely produced in insoluble form, while sample overloading does not rule out their presence among the soluble proteins. In the case of mTdT induced at 37°C, a band of insoluble protein is still clearly visible, being greatly increased when the induction temperature is reduced from 37°C to 16°C.

To maintain correct comparison between soluble and insoluble expression in SDS gel, both fractions had identical overall volume and identical loading. Expression of Z2LO10 and TEV alone without SynIDP is shown additionally in figures S15A and S16A (added). The description of mTdT expression at 37 °C was revised and the manuscript now reads (p. 9):

However, the expression of mTdT at 37 °C is low, even as insoluble protein.

Lines 277-279: “Circular dichroism (CD) spectroscopy at 37°C (Figure 2B) showed that the SynIDPs lack defined secondary structure, as they have a negative peak at ~197 nm and a positive peak at ~215 nm that are characteristic of a random coil^{29,34}.”

Can differences in the three CD spectra be attributed to different degrees of disorder? Have the Authors applied disorder predictors to the three proteins?

Originally, we did not apply disorder predictors, but rather relied on our own set of rules as described in the introduction. Due to the reviewer request we used PONDR and the 3 SynIDPs are indeed predicted to be disordered. The results are displayed below:

It is very hard to arrive at any conclusions about the degree of disorder from CD since most of the CD ‘fingerprint rules’ were recorded and deduced for globular folded proteins.

Lines 321-323: “Given that the Urry scale is derived from measurements on SynIDPs, we hypothesized that it would provide a predictive strength for assessing the hydrophathy of the disordered proteins in this work.”

I find this sentence not clear

The manuscript now reads (p. 13):

Given that the Urry scale is derived from measurements on ELPs—a SynIDP, rather than on folded globular proteins, we hypothesized that it would provide a better predictive tool to assess the hydrophathy of the SynIDPs in this work.

Lines 363-367 “Interestingly, when using double the SynIDP-mTdT concentration and 1:500 initiator:dTTP ratio, we were able to stoichiometric incorporation”

It is not clear whether "double amount" means 26.4 μM. Further, to compare SynIDP-mTdT with the commercial enzyme, the amounts of the two enzyme should be also indicated and compared.

Yes, double amount refers to concentration (as stated). The Promega TdT concentration is given by the supplier in Units and not in M, making it very hard to compare concentrations.

Lines 429-430. “The primary costs consisting of the initial purchase of oligonucleotides (~\$1k per 430 library) and DNA sequencing.”

Not clear. The main verb is missing.

Corrected to “The primary cost consists of the initial purchase of oligonucleotides (~\$1k per library) and DNA sequencing.”

Lines 432-433: “Furthermore, this unique RCA based approach for the cloning purposes of synIDPs proved useful to evolve beneficial mutations that are otherwise extremely hard to incorporate.”

In relation to the issue of introducing mutations into the nucleotide sequences of SynIDPs, the following work could be cited: Fujii R, Kitaoka M, Hayashi K. Error-prone rolling circle amplification greatly simplifies random mutagenesis. Methods Mol Biol. 2014;1179:23-9. doi: 10.1007/978-1-4939-1053-3_2.

The suggested reference (72) was added.

Lines 443-444: “We also showed that scoring the hydrophathy of the SynIDP sequence using

*Urry hydropathy scale is useful computational tool to predict the solubility of SynIDPs.”
Could the opposite approach be taken? That is, could site-directed mutagenesis of previously selected sequences identify a sequence that optimizes the score?*

Site directed mutagenesis of previously selected sequences to enhance the solubility even further is an interesting approach but is outside the scope of this manuscript.

Line 458: “promote” instead of “promot”

Corrected to promote.

A few details

The headings of the "Results" paragraphs have a rather heterogeneous style (complete sentences with active verb; nouns with specification complements; nouns and prepositions indicating experimental purposes...). I recommend a more homogeneous style.

We have addressed these and reworded headings to a more uniform style.

Throughout the text, but especially in “Materials and Methods”, the centrifugation conditions are given in a heterogeneous way, i.e. rpm or k RCF. Again I suggest using a more homogeneous way, for example, RCF, using the number of "g".

We have addressed this and made the appropriate changes.

In some cases synthetic IDPs are referred to as synIDPs, instead of SynIDPs. Please standardize the wording if not linked to a specific meaning.

All are standardized now to be SynIDP.

The origin of materials and equipment is recorded unsystematically.

Corrected in the revised manuscript.

Reviewer #3

In their abstract they claim that “Novel DNA synthesis technology” was used. DNA microarrays were first developed in the late 1980s and have been commercially available since the early 90’s, but maybe that is not what they mean. The authors need to specify what exactly is novel and summarize their RCA/partial digest approach in a sentence or two.

This may be a misunderstanding— DNA microarrays supply the starting oligonucleotide material in our approach, while the novel DNA synthesis technology refers the overall process of this paper. Although oligonucleotides from DNA microarrays have been used as starting material for overlap-based gene synthesis of nonrepetitive double stranded DNA fragments, we have demonstrated a new approach for pooled library synthesis of SynIDP genes that combines microarray oligonucleotides and RCA to generate DNA products that are used to express a diverse library of synthetic IDPs.

Is it known if the methylated C is influencing RCA efficiency or cloning in any significant way? Table S1 does not show oligonucleotides as it says in the caption but peptides. The actual DNA sequences from the oligonucleotide library should be added as the choice of codons might be important.

The incorporation of methylated C inhibits activity of the SexAI restriction enzyme, but it is not known if it alters RCA efficiency.

Table S2 was added and the title of Table S1 was changed. The manuscript in SI now reads:

Table S1: (PGX₁X₂X₃X₄)₄ Motifs encoded by the 1020 microarray-synthesized oligonucleotides.

Table S2: Designed nucleotide sequences of 1020 microarray-synthesized oligonucleotides

Font sizes in figures are sometimes too small (e.g. Fig 1 A, C) and the gel in 1A is too small to recognize anything. Is that just a space holder for 1B? Maybe indicate (with an arrow or such) that the bands were physically cut out of the gel and extracted.

Text size in Fig1A was increased. The gel shown on the bottom of Fig1A is referring to the same gel in Fig 1B and hence was replaced to make it clearer. The ladder size is shown and an arrow indicating the size of interest was added.

Enlarged image was added to supplementary information as Figure S2. The caption of Figure 1 was revised and now reads as follows:

Figure 1: The steps for identification of highly soluble SynIDPs. **A.** Generation of SynIDP gene library. A pool of 1020 ssDNA 72-nt long oligonucleotides were circularized by CirLigase and amplified using RCA. Substitution of dCTP with 10% 5-methyl-dCTP randomly incorporates 5mC into the RCA product, resulting in various DNA size products upon digestion by SexAI (Figure S1). The products can be separated by electrophoresis. Fragments of the desired size (360-576 bp) are then gel purified out and ligated into a plasmid. **B.** Restriction digest of RCA

products with 0% 5mCTP (left) vs. 10% 5mCTP (right) using the restriction enzyme SexAI shows that partial digests occur in factors of 72 bp when 5mC is incorporated. **Desired size (360-576 bp, marked with arrow) DNA are then gel purified out** **C.** Illumina Miseq analysis of cloned library of plasmid DNA verifies the presence of 865 unique motifs **(See Figure S2 for enlarged version)**. **D.** CoFi protein expression (**Figure S3**) allows identification of *E. coli* colonies that express soluble protein by an anti-His₆ western blot (left panel). Colonies were identified manually on a greyscale image (middle panel) and a black and white pixel-thresholded image overlaying the green channel (right panel), was printed and aligned underneath the plate of colonies as a visual aid for colony picking. Images represent a subset of the total set of agar plates that were analyzed (Figure S4). **E.** Dot blot of insoluble and soluble lysate fractions to determine target protein solubility. The dot blot membrane was subjected to His₆-tag antibody detection. The distinct dots marked by arrows represent an example target protein, [GTHGTP]₂₄ that was determined to be soluble. Solubility was determined by the intensity of the soluble fraction relative to the intensity of the insoluble fraction in both dot blot and PAGE gels (Figure S6, Figure S7).

Table 1 could be compressed in size; the values of the Urry scale should be restricted to significant digits.

Table 1 was revised as suggested and now presented as follows:

Table 1: Summary of the solubility of SynIDPs and **SynIDP**-mTdT fusion proteins

SynIDP status	mTdT fusion status	Sequence	Urry scale Hydropathy
insoluble		[GYGYAP] ₂₄	12.4
insoluble		[GYGHAP] ₂₈	27.3
insoluble		[GYVGTP] ₂₄	28.8
insoluble		[GYMGKP] ₂₄	33.6
insoluble		[GGMLAPGGMLAPGGMLAPGGMFAP] ₆	35.5
insoluble		[GHIGVP] ₂₄	36.9
insoluble		[GGMLAP] ₂₄	37.5
insoluble		[GGLHTP] ₂₈	41.
Soluble		[GAGAIPGAGAIPGAGAIPVAGAIP] ₆	42.3
Soluble	insoluble	[GANMPQGASIPPGANIPPGASIPP] ₆	43.2
Soluble		[GKFGTP] ₂₄	44.6
Soluble		[GLSGSP] ₂₈	45.6
Soluble		[GNGNVP] ₂₄	45.8
Soluble	insoluble	[GIGQAP] ₂₈	46.3
Soluble	soluble	[GQSGLP] ₂₄	47.0
Soluble		[GIRGNPRLRGTPPEYPGTPGIRGTP] ₇	48.1
Soluble	soluble	[GTHGTP] ₂₄	49.2
Soluble	soluble	[GAGAIPGAEAIPGAGAIPGAGAIP] ₆	49.3

In the summary or in a figure, the best tags and their DNA sequences could be highlighted for readers seeking to adapt the SynIDPs for their own research.

Best tags are displayed in Table S5 in the SI. We added the DNA sequence of the tags to that table, which now presented as follows:

Table S5: Sequences of SynIDPs with theoretical molecular weight and isoelectric points.

ID P	sequence	Mw [Da]	pI	DNA sequence
1	[GQSGLP] ₂₄	15126	5.5	ATGAGCAAAGGACCAGGTCAAAGTGGACTCCCAGGACAATCTGGCTTACC CGGACAGAGCGGTCTTCCAGGGCAATCAGGCTTACCAGGTCAAAGTGGAC TCCCAGGACAATCTGGCTTACCCGGACAGAGCGGTCTTCCAGGGCAATCA GGCTTACCAGGTCAAAGTGGACTCCCAGGACAATCTGGCTTACCCGGACA GAGCGGTCTTCCAGGGCAATCAGGCTTACCAGGTCAAAGTGGACTCCCAG GACAATCTGGCTTACCCGGACAGAGCGGTCTTCCAGGGCAATCAGGCTTA CCAGGTCAAAGTGGACTCCCAGGACAATCTGGCTTACCCGGACAGAGCGG TCTTCCAGGGCAATCAGGCTTACCAGGTCAAAGTGGACTCCCAGGACAATC TGGCTTACCCGGACAGAGCGGTCTTCCAGGGCAATCAGGCTTACCAGGTG AAAACCTGTATTTTCAGGGCCATCACCATCACCATCACGGCTAATGATGA
2	[GTHGTP] ₂₄	15390	6.7	ATGAGCAAAGGACCAGGTACACATGGCACTCCAGGAACTCACGGTACTCC GGGTACGCATGGAACCCCTGGGACTCATGGTACACCAGGTACACATGGCA CTCCAGGAACTCACGGTACTCCGGGTACGCATGGAACCCCTGGGACTCAT GGTACACCAGGTACACATGGCACTCCAGGAACTCACGGTACTCCGGGTAC GCATGGAACCCCTGGGACTCATGGTACACCAGGTACACATGGCACTCCAG GAACTCACGGTACTCCGGGTACGCATGGAACCCCTGGGACTCATGGTACA CCAGGTACACATGGCACTCCAGGAACTCACGGTACTCCGGGTACGCATGG AACCCCTGGGACTCATGGTACACCAGGTACACATGGCACTCCAGGAACTCA CGGTACTCCGGGTACGCATGGAACCCCTGGGACTCATGGTACACCAGGTG AAAACCTGTATTTTCAGGGCCATCACCATCACCATCACGGCTAATGATGA
3	[GAGAIP] ₂₄ *	13805	5.5	ATGAGCAAAGGACCAGGTGCCGGTGCAATCCCAGGGGCTGAAGCTATTCC AGGAGCGGGAGCCATACCCGGAGCAGGGCGCAATACCAGGTGCCGGTGCA ATCCCAGGGGCTGAAGCTATTCCAGGAGCGGGAGCCATACCCGGAGCAG GCGCAATACCAGGTGCCGGTGCAATCCCAGGGGCTGAAGCTATTCCAGGA GCGGGAGCCATACCCGGAGCAGGGCGCAATACCAGGTGCCGGTGCAATCC CAGGGGCTGAAGCTATTCCAGGAGCGGGAGCCATACCCGGAGCAGGGCGC AATACCAGGTGCCGGTGCAATCCCAGGGGCTGAAGCTATTCCAGGAGCGG GAGCCATACCCGGAGCAGGGCGCAATACCAGGTGCCGGTGCAATCCCAGG GGCTGAAGCTATTCCAGGAGCGGGAGCCATACCCGGAGCAGGGCGCAATAC CAGGTGAAAACCTGTATTTTCAGGGCCATCACCATCACCATCACGGCTAAT GATGA

*SynIDP-3 has a missense mutation from G (3rd amino acid) to E, every four repeats.

Reviewer #4

First, the most critical point is that no comparison data on target protein expression was provided with the other commonly used solubility-enhancing fusion tags, such as MBP, NusA, or SUMO. In addition, the authors justified the need for the design of synIDPs by mentioning that protein expression by using conventional fusion tags can result in soluble but inactive forms of proteins (lines 28-31). This justification would be valid only when the synIDPs can promote the functional folding of a target protein that is otherwise inactive even with a fusion of conventional tags. By demonstrating an improved performance of the novel synIDP fusion tags compared to those of the conventional fusion tags, the usefulness (or powerfulness) of the synIDPs may be recognized indisputably.

We thank the reviewer for the thoughtful suggestion, which we carried out as suggested. In short, MBP-mTdT and SUMO-mTdT were expressed and the enzymatic activity of the purified protein was compared to that of SynIDP1-mTdT. We also expressed SynIDP1-mTdT for cleavage via SynIDP1-TEV to demonstrate improved folding of mTdT facilitated by fusion to our SynIDPs.

The results indicate that although MBP and SUMO are able to produce soluble mTdT when fused on the N-terminus of the protein, their contributions to solubilizing their fusion partner in this case is less than that of SynIDP-1 (Figure 3D). Additionally, when testing the activity of SynIDP1-mTdT compared to that of SUMO-mTdT and MBP-TdT, the degree of polymerization of a poly T₅₀ initiator is higher for SynIDP-1-mTdT and the polydispersity is lower compared to other solubility tag mTdT fusions. Cleaved mTdT also demonstrated superior activity to that of recombinant mTdT, indicating that fusion to SynIDP1 promotes proper folding of the enzyme compared to the mTdT alone.

This data was incorporated into the manuscript, which now reads (p. 15):

To compare the performance of SynIDP with existing solubility tags, we expressed His₆-MBP-His₆-mTdT and His₆-SUMO-His₆-mTdT under the same conditions. Visualization of soluble and insoluble fractions of SynIDP-1-mTdT, SUMO-mTdT and MBP-mTdT (Figure 3D) on SDS-PAGE demonstrates significant enrichment of the target protein in the soluble fraction of SynIDP-1-mTdT compared to soluble fractions of SUMO-mTdT and MBP-mTdT. SUMO-mTdT and MBP-mTdT also exhibited a larger fraction of insoluble mTdT expression than SynIDP-1-mTdT, highlighting the improvement in soluble expression of mTdT when fused to SynIDP-1 compared to SUMO and MBP. In addition to the constructs described above, a cleaved mTdT was generated via incubation of the soluble fractions of SynIDP-1-mTdT with SynIDP-1-TEV (further described in the following sections) at 4 °C overnight. The constructs were purified using IMAC (Figure S14A, B) and SEC.

Enzymatic activity of the mTdT variants was measured by TcEP of ss-DNA (Figure 3E, Figure S14D). To that end, all mTdT variants were buffer exchanged into the optimal reaction buffer (50 mM Potassium Phosphate, 100 mM NaCl, 1 mM 2-mercaptoethanol, 0.1% Tween 20, 50% glycerol, pH 6.4) to a final concentration of 0.5 μM (Figure S14C) or 1 μM (labeled as 1X or 2X). Recombinant, soluble mTdT demonstrated very little activity (Figure S14D) as mentioned above. In contrast, all of the mTdT variants expressed as fusions to solubility tags, as well as the cleaved mTdT from SynIDP-1-mTdT demonstrated a marked improvement in nucleotide addition compared to recombinant mTdT. This increase in enzymatic activity is also interesting to note that removal of the SynIDP-1 tag from mTdT results in loss of protein activity. Further analysis of the TcEP products was performed by image processing of the original gel (Figure S14D, Figure 3E) to obtain the fluorescence intensity distribution for each of the mTdT variants (Figure 3F) as function of distance from the gel bottom. For the TcEP reaction, we were interested in obtaining a high degree of polymerization, which is indicated by a higher distance from the bottom of the gel, and low dispersity, which can be evaluated by the width of the fluorescence signal. Comparing the fluorescence intensity curves of the mTdT variants shows that SynIDP-mTdT outperforms the other mTdT variants; SynIDP1-mTdT shows the highest degree of polymerization and the lowest dispersity (Figure 3E, 3F), indicating the superior folding of mTdT within the cohort of solubility tags tested.

Figure 3. Fusion to SynIDPs rescues soluble and functional expression of mTdT. **A.** Western blot of insoluble and soluble fractions of mTdT and SynIDP-1/2/3-mTdT using anti-TdT antibody. **B.** Purified SynIDP-1/2/3-mTdT after IMAC and SEC visualized on SDS-PAGE. **C.** mTdT activity assay showing elongation of Cy5-poly-T₅₀ initiator on TBE-Urea PAGE gels. Low Range ssRNA ladder was used to quantify nucleotide addition. From left to right; ladder, initiator (negative control), Promega-TdT (positive control), SynIDP-1-mTdT 1X and 2X, SynIDP-2-mTdT 1X and 2X, SynIDP-3-mTdT 1X and 2X. **D.** Insoluble (Ins) and soluble (S) fractions of mTdT, SynIDP-1-mTdT, SUMO-mTdT and MBP-mTdT visualized on SDS-PAGE. Large increase in soluble expression of mTdT are observed when fused to SynIDP-1 compared to fusion with SUMO or MBP. Arrows indicate the desired protein product. **E.** TdT activity assay showing elongation of Cy5-poly-T₅₀ initiator on TBE-Urea PAGE gels. Low Range ssRNA ladder was used to quantify nucleotide addition. From left to right; cleaved mTdT, SynIDP-1-mTdT, SUMO-mTdT, MBP-mTdT, Ladder. Image was processed from Figure S14D (see experimental). **F.** ImageJ analysis of fluorescence intensity as measurement for dispersity and degree of polymerization of mTdT variants' reaction products. The peak at a distance of ~100 pixels stems from the dye front and is not a real product. SynIDP-1-mTdT outperforms the other variants by displaying lower dispersity and a higher degree of polymerization. The calibration of nt size to the distance on the gel (shown in the line above) was obtained by similar image analysis of the ladder.

And in Supplementary Information p55:

Figure S14: Purification of cleaved mTdT, SUMO-mTdT and MBP-TdT. (A) MBP-mTdT and SUMO-mTdT purification by IMAC visualized on SDS-PAGE. We note that MBP-His₆-mTdT and SUMO-His₆-mTdT (no His₆ tag at the terminus) were impossible to purify using IMAC (B) Soluble fractions from SynIDP-1-mTdT and SynIDP-1-TEV were combined in a 1:1 v:v ratio and incubated at 4 °C overnight. The resultant mixture was purified via IMAC and visualized on SDS-PAGE. The desired products indicated by arrows. Ins = insoluble fraction, S = soluble fraction, FT = flow through, W1 = wash with Lys buffer + 25 mM imidazole, W2 = wash with Lys buffer + 50 mM imidazole, E1 = elution with Lys buffer + 100 mM imidazole, E2 = elution with Lys buffer + 500 mM imidazole. (C) Purified mTdT variants after IMAC and SEC. (D) TdT activity assay showing elongation of Cy5-poly-T₅₀ initiator on TBE-Urea PAGE gels. Low Range ssRNA ladder was used to quantify nucleotide addition. From left to right; SynIDP-1-mTdT 2X and 1X, recombinant m-TdT 2X and 1X, SUMO-mTdT 2X and 1X, MBP-mTdT 2X and 1X, cleaved mTdT 2X and 1X, NEB-TdT (positive control), initiator (negative control), ladder. The assay indicates that recombinant mTdT has very little activity. On the contrary the cleaved mTdT as well as all of the mTdT variants expressed using solubility tag retain enzymatic activity, with SynIDP-1-mTdT demonstrating the highest processivity and lowest dispersity (Figure 3F).

We have also included a section describing our methods for the generation of cleaved mTdT, which reads as follows:

mTdT cleavage assay

SynIDP-1-mTdT and SynIDP-1-TEV were expressed as described above. The soluble fractions of the two proteins were mixed in a 1:1 v:v ratio and incubated at 4 °C overnight. The resultant

mixture was purified via IMAC-purification, as described above, and visualized on an SDS-PAGE gel. The elution fraction harboring cleaved mTdT was further purified via SEC, as described above, to separate the desired product from other contaminants (SynIDP-1-TEV, SynIDP-1).

Second, the originality of the study is overclaimed. It has already been documented that IDP tags can promote the soluble expression of recombinant proteins in at least four different papers [1-4]. However, the authors claim that they 'hypothesized' such an effect of IDP tags, without any rationale (lines 32-33 and 52-53). In addition, they cited the reference [1] as the only one previous paper that explored and used IDPs as a tag to promote soluble protein expression (lines 448-450). However, as pointed out, there are at least three more papers that demonstrated the use of IDP as a solubility-enhancing fusion tag, of which one paper [3] is already included in the reference list in the manuscript.

References

[1] *Biochemistry* 51, 7250–7262 (2012) (reference 23 in the manuscript)

[2] *Biomacromolecules* 15, 1194–1203 (2014)

[3] *Appl. Environ. Microbiol.* 88, e0009722 (2022) (reference 22 in the manuscript)

[4] *J. Microbiol.* 60, 960–967 (2022)

We apologize for not providing the rationale for the hypothesis. The corresponding section in the manuscript was revised and now reads as follows (p. 2):

It has been previously shown that disordered regions at the N terminus of folded native proteins promotes solubility and prevents aggregation of the protein^{13,14}. Due to lack of secondary structure, these disordered regions have high conformational freedom and hence believed to act as entropic bristles, preventing aggregation. Based on this knowledge, we hypothesized that polypeptide tags derived from intrinsically disordered proteins (IDPs) can promote the soluble expression of recombinant proteins.

Native IDPs are highly dynamic, as they have significant stretches of amino acids that lack a defined tertiary structure and possess a high degree of solvent-exposure^{22,23}. These properties of IDPs could enable folding of their fused protein partner, and thereby promote soluble protein expression²⁴⁻²⁶.

We thank the reviewer for providing us with the 2 additional refs that we have overlooked. They are now incorporated in the discussion (p. 20):

We note that this is a relatively new and interesting application of IDPs, as there is little previously published work that explored and used IDPs as tags to promote soluble protein expression^{24,25,89}

-Lines 56-58: The second reason why 'synthetic' IDPs, instead of native IDPs, are required is less convincing. Because recombinant proteins are mostly expressed in a heterologous host, potential interference of a native IDP with cellular function or metabolism would not be a problem. Conversely, a synthetic IDP might have a potential adverse effect on the cellular function or metabolism of the host.

Some native IDPs that are from another host organism can be toxic to *E. coli*, for example as described in Ref 84 (doi.org/10.1016/j.bbrc.2006.01.151). More broadly, native IDPs have a specific cellular function, some of which is dictated by their folded domains, which are unnecessary for the goal of this paper—the soluble expression of proteins. We hypothesized that the unstructured domain of native IDPs—the intrinsically disordered regions that connect the

folded domains of native IDPs— are responsible for promoting the soluble expression of native IDP fusions. We test this hypothesis in this paper by identifying SynIDPs that are completely disordered and have no putative biological function so that they are likely to be non-toxic. Their lack of toxicity is suggested by the fact that were designed from PX_nG motifs, which we have shown in multiple studies have no toxicity in vivo; parenthetically —ELPs—the parental SynIDPs with a VPGXFG motif have shown no toxicity in humans in multiple clinical trials. Their lack of toxicity, lack of designed biologic function, and the fact that they do not interfere with the function of the fused protein suggests a new application that we allude to in the discussion section of the paper for the future use of these SynIDP fusion in vivo, where the SynIDP could improve the pharmacokinetics of biologic drugs they are fused to. Because these SynIDPs are highly soluble (and hence highly solvated) we hypothesize that they will yield a long plasma half-life in vivo, a hypothesis that we will test in in a follow-on study.

-Line 58: The term 'SynIDPs' was already mentioned in line 34.

We edited the sentence, which now reads (p. 2):

We report the *de novo* design of small (<20 kDa) and highly soluble SynIDPs that confer solubility...

-Lines 255 and 334: Which one is correct, SynIDP-tev-H6 or SynIDP-H6-tev?

SynIDP-tev-H6 is the correct one. We made sure it is consistent through the manuscript.

-Figure S7: The Western blot data on murine TdT (mTdT) expression is weird. In the lysate, the sizes of mTdT fused to the SynIDPs are all around 73 kDa, but in the soluble fraction, the sizes are shifted to around 59 kDa. It cannot be due to the cleavage of SynIDPs from the full-length proteins because, in that case, the 59 kDa bands must also appear in the lysate with the same band intensities observed in the soluble fraction. I cannot understand this result, but there is no explanation for this important issue.

We mislabeled the Mw on the ladder and apologize for that. Figure S9 was corrected.

-Figure S8: The band position of SynIDP-3 on the gel (above 20 kDa marker protein) was shifted downward (below 20 kDa and close to 15 kDa) after SEC. Please provide any explanation for this result.

We thank the reviewer for pointing this out. This occurred due to mislabeling the Mw on the ladder and we apologize for that. The Figure has been corrected.

-Figure 3A: The band positions of mTdT proteins are incorrect. The untagged mTdT should appear around 59 kDa.

We apologize for the mislabeled ladder. That was fixed in Fig 3A as well.

-Throughout the manuscript, the authors claimed that the target proteins were functional without cleaving off the SynIDP tags. This might be beneficial in terms of production cost. However, during IMAC (Figure S11A), most of the recombinant proteins appeared in the flow-through fraction, which indicates inefficient binding of the proteins to the Ni-NTA resin. The tethered SynIDPs might interfere with the binding. In this situation, even though the SynIDPs could improve the soluble expression of the target protein, the final yield of purified protein would be very low, severely reducing the cost-effectiveness of protein production. In this regard, the cleavage of SynIDP tags seems necessary to improve the final yield of the target protein. In addition, although the TEV cleavage site was inserted in the constructs with SynIDPs, it has never been used (cleaved) in this study! Therefore, I suggest further experiments on the cleavage of SynIDP tag from SynIDP-tev-H6-mTdT prior to the protein purification by IMAC to see whether the tag-cleaved H6-mTdT can be highly purified without any binding interference from the SynIDP tags. For this purpose, I recommend the use of both commercially available TEV protease and SynIDP-TEV protease constructed in this study to compare the performances on the same substrate.

The architecture of the SynIDP fusions could be improved upon by adding His tag to another location of the protein, such as the C-terminus, to improve His-tag accessibility. However, in the process of generating the IDPs, we used an anti-His6 antibody in a western-blot to identify soluble expression of SynIDP harboring colonies, which was crucial to identification of SynIDP-1,2,3. Thus, all further studies were done with this architecture as SynIDP-tev-His architecture was what we originally screened for.

We have carried out additional studies to demonstrate that we can cleave our SynIDP1-mTdT with SynIDP-TEV and purify the cleaved mTdT product in a soluble form. Additionally, this mTdT demonstrates superior activity to that of recombinant mTdT indicating that the protein can maintain function after cleavage from the solubility tag.

In addition to the new data added to the manuscript and displayed above, a comment was added (page 14):

We note that the efficiency of IMAC purification could be improved by moving the His₆ tag to the C-terminus of the fusion, which should improve its accessibility to the Ni-NTA ligand on the IMAC resin.

-Figure S12B: For SynIDP-1-Z2LO10, the two lanes for W2 and E seem to be mislabeled to each other.

W2 and E were switched, and has been corrected.

-Lines 452-455: The authors claimed that they expressed and purified active mTdT using a fusion tag for the first time. Regardless of whether it is really the first study, I do not understand why this is important. As the authors have used in this study, recombinant TdTs from calf thymus are already commercially available. The activity of SynIDP-mTdT seems even lower than that of the commercially available enzyme (lines 363-366). Are there any beneficial characteristics of mTdT that are distinct from other TdT isozymes?

TdT has been used as a homopolymer tailing for cDNA synthesis and the TUNEL apoptosis assay for decades. In the past decade, we have pioneered the use of TdT to make long and monodisperse polynucleotides that self-assemble into nanoparticles for application in drug delivery (doi.org/10.1021/acs.biomac.3c008880). In the past five years, another exciting application of TdT has emerged for the de novo enzymatic synthesis of sequence-defined DNA oligonucleotides. This new technology has generated significant commercial interest with the formation of many start-ups world-wide. Developing new methods for the synthesis of TdT is hence of great interest for many applications. Unfortunately, TdT from commercial vendors is very expensive and prohibits the scale-up of many of these applications.

When we examined commercially available TdT (which we hypothesize is recombinant bovine-TdT), Promega TdT performed the best among all vendors. Our attempts to express soluble recombinant bovine TdT were not successful (although we tried following the published patent (US 7,494,797 B2)). Upon expressing m-TdT, there was very little expression as soluble and active protein. Fusing mTdT with MBP helped with solubility and activity to some degree, but the enzyme has high dispersity and low processivity. In addition, we also compared our SynIDP-mTdT with another solubility tag, SUMO, which worked better than MBP but the SynIDP fusion was still superior in terms of processivity and dispersity of the DNA product. Being able to produce highly processive TdT at far lower cost than purchasing the enzyme, using genes that will be deposited in Addgene and protein expression and purification protocols that will be freely available to researchers world-side will democratize the use of this important biotechnology enzyme. The fact that we state that this is the first example of a TdT fusion that is soluble and active is interesting and relevant because it shows that the tag does not have to be cleaved prior to use of the enzyme, which will further lower the complexity and cost of producing this industrially important enzyme.

REVIEWER COMMENTS

Reviewer #1 (Remarks to the Author):

Thank you for your detailed response to my queries. I find the responses satisfactory and approve the manuscript for publication.

I hope your findings will be useful to the protein expression community. All the best.

Reviewer #2 (Remarks to the Author):

The Authors have addressed most of the requests made by myself and other Reviewers, resulting in a significant improvement in the manuscript quality. However, I require clarification on some points that I raised during the previous round of revisions, and some others that were elicited by the revised version of the manuscript. The page numbers I am referring to correspond to those in the PDF document with track changes.

- The Authors were asked to clarify/discuss the rationale for their choice of SynIDP tag size. In their rebuttal letter, the Authors presented the main reasons for their choice: 1) Tags smaller than 10 kDa are poorly expressed, making them difficult to characterize. 2) Tags larger than 20 kDa can cause metabolic stress in cells overexpressing fusion proteins. 3) A previous study (Santner et al., 2012) identified a 15 kDa disordered tag as one of the most effective. This size seems suitable to provide the function of an entropic bristle for target proteins chosen from Tang and co-authors, which have a molecular weight ranging from ca. 27 kDa (TEV) to 58 kDa (mTdT).

In my opinion, these points rationally motivate the experimental design and, in particular, the selection of nucleotide sequences of 360-576 bp, which encode tags of 120-192 aa and, indeed, gave rise to SynIDP-1 (15.1 kDa), SynIDP-2 (15.4 kDa) and SynIDP-3 (13.8 kDa).

Authors are now encouraged to include these reasons in the main text. I apologize if this request was not stated clearly enough in my previous report.

- On page 7, lines 192-193, the sentence "Several rounds of expression were carried out to accommodate multiple backup clones" is unclear. Please rephrase it or add further explanation.

- On page 10, lines 284-286, the sentence on protein condensation is not yet clear ("No increase in absorbance was detected over this temperature range, indicating a lack of LCST phase separation under these conditions and ensures that no coacervation is expected due to elevated temperature or salt concentration"). I would say in clearer way that the experimental conditions applied do not support

protein condensation, as indicated by the absence of LCST phase separation and the lack of absorbance increase over the temperature range they tested.

- On page 13, line 330, a hyphen is probably missing at the end of the sentence.

- On page 14, line 370, regarding the TdT activity assay. The Authors were asked to indicate the amount of commercial enzyme used as a positive control for the TdT activity assay, in order to substantiate the comparison with the activity of their recombinant mTdT. The Authors claimed that the concentrations of the commercial and recombinant mTdT enzymes are not homogeneously expressed as they are presented in terms of U and molarity, respectively. However, to ensure reproducibility of the experimental data, the Authors should report at least the number of units of commercial enzyme used in the assays. On page 29, line 805 (Material and Methods), I guess that they have reported the concentration of commercial mixtures (1.2 U/ μ L). Please clarify or add information.

- New experiments aimed at comparing SynIDPs with SUMO- and MBP-based solubility tags were requested by other Reviewers and are partly represented by new panels D and E of Figure 3.

1) Panel D: SDS-PAGE analysis of the soluble and insoluble protein fractions of mTdT fused with SynIDP, SUMO and MBP displays unbalanced loading of cell extracts expressing different fusion proteins. In particular, involuntarily loading of a smaller quantity of insoluble proteins in the SynIDP sample risks undermining the objectivity of the results, artificially supporting the hypothesis of greater effectiveness of this tag.

2) Panel E: PAGE shows a ss-RNA ladder without any size labelling.

3) Overall, the quality of “old” panels remains poor (too small letters used for labelling, and grainy images).

- It has been noted by other Reviewers that the effect of the SynIDP tag is unclear due to the undefined distribution of the three target proteins in the soluble and insoluble protein fractions, in the absence of a tag (tag-free forms). The Authors reply that the requested experimental evidence can be found in the SDS-PAGEs of supplementary figures S8, S15 and S16. It should be noted that these figures are illegible due to the overload of soluble protein samples, at the point that is very hard to distinguish the presence of a specific target protein. Although evidence can be drawn for the insolubility of mTdT (see new panel D of figure 3 and S8B) and TEV (see panel S16A), experimental data are still missing to confirm the insolubility of tag-free Z2LO10. I suggest that the solubility data of the tag-free TEV and Z2LO10 proteins be reported in a single panel to be included in Figure 4.

- On page 20, lines 510-513, the Authors speculate that “SynIDPs with a high positive net charge may complex with other positively charged macromolecules, such as DNA, and precipitate out, despite having high hydrophathy”. The sentence contains an error in attributing positive charges to DNA, which may be a simple oversight. Furthermore, it is somewhat misleading to attribute the low tendency of SynIDPs to precipitate to their high hydrophathy. This requires further consideration: although the hydrophathy index of Urry assigns higher values to less hydrophobic molecules, it would be clearer to simply refer to the low hydrophobicity of SynIDPs.

Reviewer #3 (Remarks to the Author):

The authors have addressed my concerns.

Reviewer #4 (Remarks to the Author):

The authors have appropriately answered all my concerns and demonstrated the usefulness of the synIDP tag by providing additional data.

Reviewer comments are provided below in *italics*, and our responses are shown below each comment in blue. Manuscript text is shown in Arial font with specific sections added being highlighted.

We thank reviewer #2 for their careful attention to the details, which has helped improve the manuscript.

Reviewer #2 (Remarks to the Author):

- The Authors were asked to clarify/discuss the rationale for their choice of SynIDP tag size. In their rebuttal letter, the Authors presented the main reasons for their choice: 1) Tags smaller than 10 kDa are poorly expressed, making them difficult to characterize. 2) Tags larger than 20 kDa can cause metabolic stress in cells overexpressing fusion proteins. 3) A previous study (Santner et al., 2012) identified a 15 kDa disordered tag as one of the most effective. This size seems suitable to provide the function of an entropic bristle for target proteins chosen from Tang and co-authors, which have a molecular weight ranging from ca. 27 kDa (TEV) to 58 kDa (mTdT).

In my opinion, these points rationally motivate the experimental design and, in particular, the selection of nucleotide sequences of 360-576 bp, which encode tags of 120-192 aa and, indeed, gave rise to SynIDP-1 (15.1 kDa), SynIDP-2 (15.4 kDa) and SynIDP-3 (13.8 kDa). Authors are now encouraged to include these reasons in the main text. I apologize if this request was not stated clearly enough in my previous report.

The rationale for the specific size of the solubility tag was added to the manuscript and the manuscript now reads (page 3):

Hence, we decided to create SynIDPs in the size range of 10-20 kDa that are completely disordered, motivated by the hypothesis that a complete lack of secondary structure would impart the highest possible degree of solvent exposure, and is also unlikely to impart—potentially interfering—biological function to these SynIDPs. We chose SynIDPs in the 10-20 kDa size range, as in our experience, polypeptides smaller than 10 kDa are poorly expressed, making them difficult to characterize, while those larger than 20 kDa can cause metabolic stress in cells when they are fused to other proteins and expressed from a strong promoter. Furthermore, a previous study exploring disordered proteins as solubility tags identified a 15 kDa disordered tag as one of the most effective²⁵. We hypothesized that these SynIDPs may promote the solubility of proteins that they are fused to, and therefore may serve as a useful tag for the soluble expression of proteins that are known to form inclusion bodies in *E. coli*.

- On page 7, lines 192-193, the sentence “Several rounds of expression were carried out to accommodate multiple backup clones” is unclear. Please rephrase it or add further explanation.

This line describes a technical detail - Duetz Microflask systems allows simultaneous growth of 48 genes, and since we had more than 48 genes, more than one flask was required. As the reviewer found it confusing, we deleted this line from the manuscript.

- On page 10, lines 284-286, the sentence on protein condensation is not yet clear (“No increase in absorbance was detected over this temperature range, indicating a lack of LCST phase separation under these conditions and ensures that no coacervation is expected due to

elevated temperature or salt concentration”). I would say in clearer way that the experimental conditions applied do not support protein condensation, as indicated by the absence of LCST phase separation and the lack of absorbance increase over the temperature range they tested.

To make the statement about the lack of LCST more clear, the text on page 10 was changed, and the manuscript now reads:

As native IDPs are frequently known to have lower/upper critical solution temperature phase behavior (LCST/UCST) that causes their phase separation upon heating or cooling respectively^{1, 2, 3} and because these SynIDPs were identified from a library of PX_nG motifs that can exhibit similar LCST or UCST phase transition⁴, we carried out thermally ramped turbidity experiments in PBS and 1 M NaCl in a temperature range of 15-80 °C (Figure S12). No increase in absorbance was detected over this temperature range, indicating a lack of LCST or UCST phase separation under these conditions. These results suggest that SynIDP fusions are not likely to phase separate under the conditions likely to be encountered during their purification.

- On page 13, line 330, a hyphen is probably missing at the end of the sentence. Hyphen was added.

- On page 14, line 370, regarding the TdT activity assay. The Authors were asked to indicate the amount of commercial enzyme used as a positive control for the TdT activity assay, in order to substantiate the comparison with the activity of their recombinant mTdT. The Authors claimed that the concentrations of the commercial and recombinant mTdT enzymes are not homogeneously expressed as they are presented in terms of U and molarity, respectively. However, to ensure reproducibility of the experimental data, the Authors should report at least the number of units of commercial enzyme used in the assays. On page 29, line 805 (Material and Methods), I guess that they have reported the concentration of commercial mixtures (1.2 U/μL). Please clarify or add information.

The working concentration of the commercial TdT enzymes was 1.2 U/μL, per manufacturer instructions.

The working concentration of the TdT expressed by us was adjusted to 0.5 or 1 μM (quantified by nanodrop).

This information was already presented on page 14 line 367 and in the methods section. We now have added the concentration to the end of page 14, beginning of page 15 as well:

as a positive control we used commercially available TdT (Promega or NEB, working concentration of 1.2 U/μL).

- New experiments aimed at comparing SynIDPs with SUMO- and MBP-based solubility tags were requested by other Reviewers and are partly represented by new panels D and E of Figure 3.

1) Panel D: SDS-PAGE analysis of the soluble and insoluble protein fractions of mTdT fused with SynIDP, SUMO and MBP displays unbalanced loading of cell extracts expressing different fusion proteins. In particular, involuntarily loading of a smaller quantity of insoluble proteins in the SynIDP sample risks undermining the objectivity of the results, artificially supporting the hypothesis of greater effectiveness of this tag.

To maintain correct comparison between soluble and insoluble expression between the different strains on SDS gel, both fractions were resuspended to the same volume prior to loading. Gels were all loaded at 5X dilution with identical volumes run. To convince the reviewer of the validity of the results and that it is not an artifact, an additional SDS-gel from another batch expression is presented below.

2) Panel E: PAGE shows a ss-RNA ladder without any size labelling.

We apologize. Ladder size labeling was added.

3) Overall, the quality of “old” panels remains poor (too small letters used for labelling, and grainy images).

The reviewer did not point out a specific figure, however we have identified panel C in Figure 1 as an image where the fonts were too small to easily read. We have addressed this and replaced the figure in the manuscript.

Figure 1 now shows:

- It has been noted by other Reviewers that the effect of the SynIDP tag is unclear due to the undefined distribution of the three target proteins in the soluble and insoluble protein fractions, in the absence of a tag (tag-free forms). The Authors reply that the requested experimental evidence can be found in the SDS-PAGEs of supplementary figures S8, S15 and S16. It should be noted that these figures are illegible due to the overload of soluble protein samples, at the point that is very hard to distinguish the presence of a specific target protein. Although evidence can be drawn for the insolubility of mTdT (see new panel D of figure 3 and S8B) and TEV (see panel S16A), experimental data are still missing to confirm the insolubility of tag-free Z2LO10. I suggest that the solubility data of the tag-free TEV and Z2LO10 proteins be reported in a single panel to be included in Figure 4.

We apologize for the insufficient quality of the gel presented in figure S8. The brightness/contrast was adjusted to better visualize the different bands.

We find that the solubility/ insolubility of a target protein **without solubility tag** is not important enough to be presented as a figure panel in the main text. As the reviewer pointed out, the insolubility of TEV is obvious in Figure S16A, and we hence chose to add a panel to Figure S15 to further illustrate the insolubility of Z2LO10. Figure S15 now shows:

Figure S15: Purification and activity of SynIDP-Z2LO10. Z2LO10 and SynIDP-Z2LO10 insoluble and soluble fraction (A) and SynIDP-Z2LO10 purification by IMAC (B) visualized on SDS-PAGE gel. T=total lysate, Ins = insoluble fraction, S = soluble fraction, FT = flow through, W2 = wash with PBS buffer + 50 mM imidazole, E = elution with PBS buffer + 100 mM imidazole, except for the SynIDP-2-Z2LO10 where we used 500 mM imidazole. (C) Log-fold dilutions of SynIDP (controls) were incubated with CT-2A-EGFRviii, an EGFR positive murine glioma cell line for 48 h and tested for viability by an MTS assay. (D) Z2LO10 total lysate and soluble fraction visualized on SDS-PAGE gel. The bands marked by an arrow in each gel indicates the location of the target proteins.

- On page 20, lines 510-513, the Authors speculate that “SynIDPs with a high positive net charge may complex with other positively charged macromolecules, such as DNA, and precipitate out, despite having high hydrophathy”. The sentence contains an error in attributing positive charges to DNA, which may be a simple oversight. Furthermore, it is somewhat misleading to attribute the low tendency of SynIDPs to precipitate to their high hydrophathy. This requires further consideration: although the hydrophathy index of Urry assigns higher values to less hydrophobic molecules, it would be clearer to simply refer to the low hydrophobicity of SynIDPs.

We corrected this mistake, and the paragraph now reads (page 20):

The most soluble synthetic IDPs also lack positive charge. We speculate that SynIDPs that have high positive net charge may complex with other **negatively** charged macromolecules such as DNA and precipitate out even though they have a high hydrophathy (**low hydrophobicity**).

1. Goncalves-Kulik M, *et al.* Low Complexity Induces Structure in Protein Regions Predicted as Intrinsically Disordered. *Biomolecules* **12**, (2022).
2. MacEwan SR, Hassouneh W, Chilkoti A. Non-chromatographic purification of recombinant elastin-like polypeptides and their fusions with peptides and proteins from Escherichia coli. *J Vis Exp*, (2014).
3. Christensen T, Hassouneh W, Trabbic-Carlson K, Chilkoti A. Predicting transition temperatures of elastin-like polypeptide fusion proteins. *Biomacromolecules* **14**, 1514-1519 (2013).
4. Quiroz FG, Chilkoti A. Sequence heuristics to encode phase behaviour in intrinsically disordered protein polymers. *Nature materials* **14**, 1164-1171 (2015).

REVIEWERS' COMMENTS

Reviewer #2 (Remarks to the Author):

The authors have answered my questions thoroughly and accurately. Their detailed answers have increased my confidence in the manuscript. I therefore recommend that it be accepted for publication.

Best wishes